# VIRTUAL COMMUNITY: AN OPEN WORLD FOR HUMANS, ROBOTS, AND SOCIETY

**Qinhong Zhou**[1*]   **Hongxin Zhang**[1*]   **Xiangye Lin**[1*]   **Zheyuan Zhang**[2,3*]
**Yutian Chen**[2,4]   **Wenjun Liu**[1]   **Zunzhe Zhang**[1]   **Sunli Chen**[1]   **Lixing Fang**[1]
**Qiushi Lyu**[2]   **Xinyu Sun**[1]   **Jincheng Yang**[2]   **Zeyuan Wang**[2]   **Bao Chi Dang**[1]
**Zhehuan Chen**[1]   **Daksha Ladia**[1]   **Quang Vinh Dang**[1]   **Jiageng Liu**[1]   **Chuang Gan**[1,2]
[1]UMass Amherst   [2]MIT-IBM Watson AI Lab   [3] Johns Hopkins University   [4]CMU

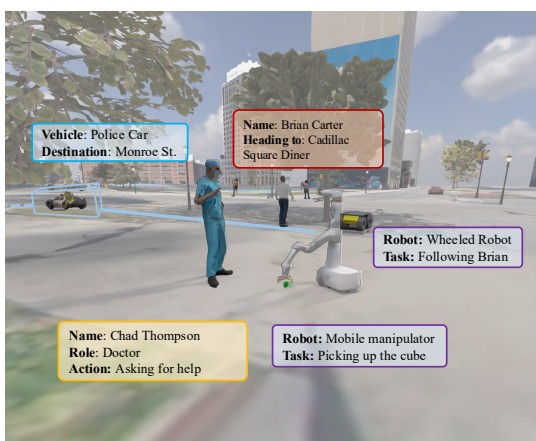
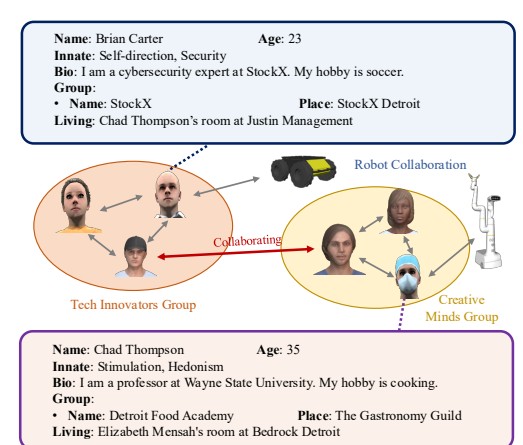

Figure 1: **Virtual Community supports embodied multi-agent simulation in open-world environments.** We introduce an automated pipeline that generates open-world scenes and agent communities, with agents instantiated as humanoid avatars or robots to enable diverse social interactions.

## ABSTRACT

The rapid progress of AI and robotics may profoundly transform society, as humans and robots begin to coexist in shared communities, bringing both opportunities and challenges. To explore this future, we present Virtual Community—an open-world platform for humans, robots, and society—built on a universal physics engine and grounded in real-world 3D scenes. With Virtual Community, we aim to enable the study of embodied social intelligence at scale. To support these, Virtual Community features: 1) An open-source multi-agent physics simulator that supports robot, human, and their interactions within a society; 2) A large-scale, real-world aligned environment generation pipeline, including vast outdoor space, diverse indoor scenes, and a community of grounded agents with rich characters and appearances. Leveraging Virtual Community, we propose two novel challenges. The *Community Planning Challenge* evaluates multi-agent reasoning and planning in open-world settings, such as cooperating to help agents with daily activities and efficiently connecting other agents. The *Community Robot Challenge* requires multiple heterogeneous robots to collaborate in solving complex open-world tasks. We evaluate various baselines and demonstrate the challenges in both high-level open-world task planning and low-level cooperation controls. We have open-sourced our project[*] and hope that Virtual Community will unlock further study of human-robot coexistence in open worlds.

## 1 INTRODUCTION

In recent years, the development of intelligent embodied agents has been propelled by advances in virtual simulators (Savva et al., 2019; Kolve et al., 2017; Anderson et al., 2018; Todorov et al.,

---

[*]Website: https://virtual-community-ai.github.io/

2012; Savva et al., 2017; Li et al., 2021; Xiang et al., 2020b; Makoviychuk et al., 2021; Gan et al., 2021; Cheng et al., 2024; Wu et al., 2024; Wang et al., 2024a; Li et al., 2024; Zhong et al., 2024; Zhuang* et al., 2025; Du et al., 2019). However, most of these platforms focus on robots (Tao et al., 2024; Xiang et al., 2020b; Li et al., 2024), human-like agents (Puig et al., 2018; 2020), or only a limited number of agents with simple interactions (Puig et al., 2023; Gan et al., 2021). In contrast, support for large, heterogeneous communities of human and robot agents in scalable open worlds remains limited. Such worlds allow agents to freely explore large, non-linear indoor–outdoor environments instead of following fixed paths or sequences of levels, yet existing platforms rarely offer this capability at scale, constraining the study of complex multi-agent behaviors between humans and robots.

To address this challenge, simulators must support the following key features. First, they should offer physically realistic simulations that accommodate large communities of human-like avatars and robots. Existing multi-agent embodied AI platforms (Puig et al., 2018; 2023; Gan et al., 2021; Li et al., 2021; Wu et al., 2024) typically handle only small groups of avatars or robots, or provide limited physics-based interactions, thereby constraining the realism of community-level behaviors. Second, the simulator must support the creation of diverse, populated worlds, including large-scale 3D environments and scene-grounded agent communities. Current approaches fall into two categories: manual design or procedural generation (Wang et al., 2024a; Gan et al., 2021; Tsoi et al., 2022; Wu et al., 2024; Gao et al., 2024), which enable rich agent–environment interactions but suffer from limited diversity and realism; and 3D reconstruction methods (Savva et al., 2019; Shen et al., 2021), which produce visually realistic and varied scenes but require extensive visual input and often yield low-interactivity environments in open-world settings.

In this paper, we present Virtual Community, an open world for humans, robots, and society. Virtual Community addresses these challenges by building a unified simulation framework for human-like agents and robot agents based on the Genesis (Authors, 2024) physics engine and integrating large-scale, real-world geospatial data with generative models to produce interactive, scalable open worlds (Figure 1). The platform offers two key advancements:

**Unified Simulation for Avatars and Robots** Virtual Community simulates human-like avatars and diverse robots within the generated open worlds using a unified framework based on the Genesis (Authors, 2024) physics engine, supporting diverse physical and social interactions among different types of agents. Virtual Community also provides robot and human agents with a unified interface with distinct observation and action spaces.

**Open World Generation from Real Scenarios** Virtual Community fully automates the generation of open worlds with several key features: (1) scalable, real-world–aligned outdoor scenes of customizable size and quantity, along with corresponding indoor scenes and annotations; and (2) generation of agent communities endowed with scene-grounded profiles and social relationship networks. Virtual Community combines generative models with real-world geospatial data, ensuring scalability in data volume, realism, and extent.

Virtual Community enables a variety of new possibilities in embodied AI research. The expansive open-world scenes and their agent communities introduce a new challenge of multi-agent task planning in open worlds. We introduce the *Community Planning* challenge as a first step in this direction. This challenge includes assistant tasks, in which human agents interact with others to provide assistance in daily open world activities, and social influence tasks, in which human agents must efficiently explore the community and connect with one another. Virtual Community also supports physically realistic simulations of interactions, for which we propose the *Community Robot* challenge. This challenge tasks robot agents with cooperating to complete tasks that involve both indoor and dynamic open-world environments.

Our simulator advances the field by enabling unified simulations of human and robot communities in generated open worlds, surpassing existing solutions in both scope and capability. By overcoming limitations in the scalable simulation of humans, robots, and societies, we pave the way for studying embodied general intelligence that can coexist with complex, interconnected human communities.

## 2 RELATED WORKS

**Embodied AI Simulation** Recently, embodied AI has seen significant advancements through the development of simulation platforms. Most existing simulators primarily focus on household tasks within indoor scenes (Beattie et al., 2016; Savva et al., 2019; Yi et al., 2018; Das et al., 2018; Xiang et al., 2020a; Shen et al., 2021; Li et al., 2021; Puig et al., 2018; Kolve et al., 2017; Yan et al., 2018; Li et al., 2024; Tao et al., 2024; Deitke et al., 2020; 2022b), while some have extended support to outdoor environments (Gan et al., 2021; Tsoi et al., 2022; Wang et al., 2024a; Dosovitskiy et al., 2017; Kendall et al., 2018; Gulino et al., 2023; Wu et al., 2024). However, existing platforms lack the diverse and scalable outdoor environments needed to support larger agent populations and more complex multi-agent interactions. In contrast, this paper introduces a simulation platform with expansive open-world environments, integrating both indoor and scalable outdoor scenes to facilitate broader agent interactions and enable more intricate task scenarios.

**Embodied Social Intelligence** Current research on *Embodied Social Intelligence* is often limited to small agent populations in constrained household scenarios (Puig et al., 2020; Zhang et al., 2023; Stone et al., 2022; Savva et al., 2019; Jain et al., 2020; Szot et al., 2023; Zhang et al., 2024) or simplified to 2D or grid worlds (Carroll et al., 2019; Suarez et al., 2019; Tsoi et al., 2020; Samvelyan et al., 2019; Yu et al., 2024; Yang et al., 2024a), hindering model development in the open world. Specifically, (Park et al., 2023) demonstrates the robust simulation of human-like agents within a symbolic community, ignoring the 3D perception and realistic physics in the open world. (Wang et al., 2023c) studies human-like simulation guided by system 1 processing with basic needs. Predominant approaches, such as multi-agent reinforcement learning (MARL) and other planning models, face several limitations when applied to open-world settings. MARL, for instance, often struggles with scalability due to the exponential growth of state and action spaces as the number of agents increases (Wen et al., 2022). This makes it difficult to learn effective policies in complex, dynamic environments. Additionally, MARL approaches typically require extensive training data and computational resources, which may not be feasible in real-world applications. Other planning models, while potentially more efficient, often lack the adaptability required to handle the unpredictable nature of open-world interactions. They may rely on predefined rules or assumptions that do not hold in all scenarios, leading to suboptimal performance and limited generalization to new contexts (Puig et al., 2020).

**Foundation and Generative models for Embodied AI** With the recent advance of foundation models (Bubeck et al., 2023; Liu et al., 2023; Driess et al., 2023; Blattmann et al., 2023), numerous works have explored how they can help build powerful embodied agents (Wang et al., 2023b; Xi et al., 2023; Sumers et al., 2023; Wang et al., 2023d; Ahn et al., 2022; Sharma et al., 2021; Wang et al., 2023a; Park et al., 2023; Hong et al., 2024; Black et al., 2024b), and scenes for simulation (Yang et al., 2024c; Hu et al., 2024; Höllein et al., 2023; Schult et al., 2023; Deitke et al., 2022a; Fu et al., 2021; Yang et al., 2024b; Feng et al., 2024; Tang et al., 2023; Paschalidou et al., 2021; Shcherbyna et al., 2024; Deitke et al., 2023). RoboGen (Wang et al., 2024c) utilizes foundation models to automatically generate diverse tasks, scenes, and training supervision, scaling up robotic skill learning with minimal human input. In contrast, our work fully integrates a generative pipeline into the simulation platform to create expansive open-world scenes and agent communities.

## 3 GENERATING OPEN WORLDS FOR SIMULATION

### 3.1 SCALABLE 3D SCENE CREATION

The existing 3D geospatial data API[*] provides extensive data in terms of quantity and diversity. However, they are not directly suitable for embodied AI research. First, these geospatial data often contain noise, including transient objects and unrealistically rugged terrain that can disrupt simulations. Second, visual quality is inadequate for ground-level agent perspectives because these environments are typically reconstructed from aerial imagery and lack visual detail at ground level.

To bridge this gap, we propose an online pipeline that performs comprehensive cleaning and enhancement in both geometry and texture to make the scenes suitable for embodied AI simulations. This pipeline consists of four steps: mesh simplification, texture refinement, object placement, and

---

[*]https://www.google.com/maps/

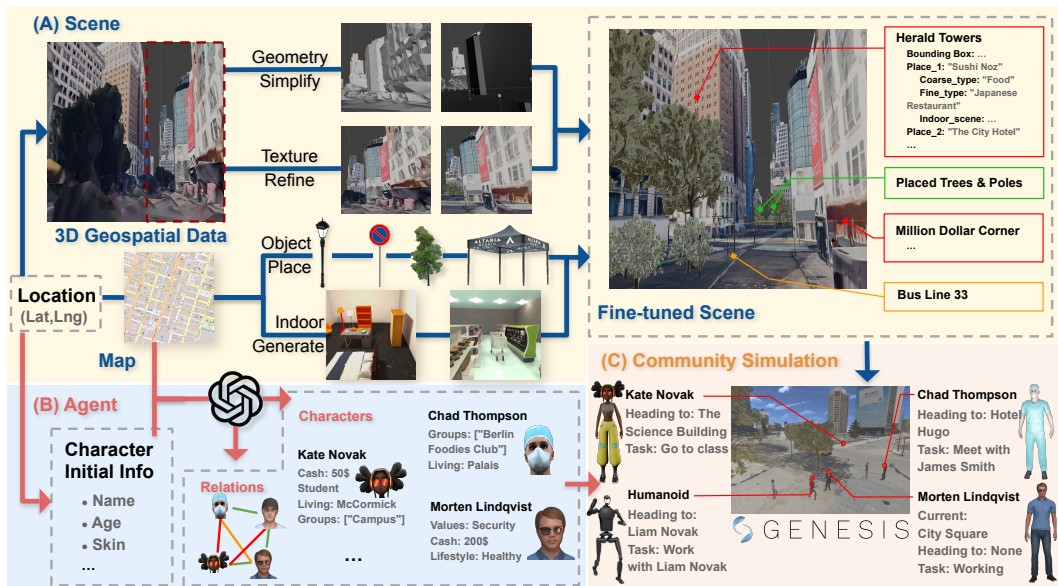

Figure 2: **Framework of the Virtual Community Generation Pipeline.** This pipeline generates scenes and corresponding human agents from real-world geospatial data. The **scene generation** component (A) refines rough 3D data by using generative models to enhance textures and geospatial data to simplify geometry. It also utilizes generative methods to create interactive objects and detailed indoor scenes. The **agent generation** component (B) leverages LLMs to generate agent characters and social relationship networks based on scene descriptions. (C) We simulate the human communities and robots in the open world scenes based on Genesis engine.

automatic annotation. The pipeline supports automatic creation of 3D urban scenes at arbitrary locations. Using this pipeline, we generated 35 annotated scenes of various cities worldwide and present some qualitative examples of these scenes in Figure 3.

**Geometry Reconstruction and Simplification** Since the mesh topologies in 3D geospatial data sources are unreliable for embodied AI simulations, we decompose scenes into terrain, building, and decorative-roof elements, then apply specialized reconstruction operations to each component to make the entire scene simulation-ready. The terrain is generated procedurally from sparse reference elevation points via bilinear interpolation. We then derive simple, topologically sound building meshes using OpenStreetMap (OSM) data. Each building mesh is automatically adjusted to better match the Google 3D Tiles geometry and to align with the terrain elevation. By aligning mesh geometries to OSM primitives, we remove unnecessary details and artifacts—such as distorted surfaces and irregular shapes resulting from aerial reconstruction errors—thereby denoising the meshes for more efficient physics simulations and improved rendering performance.

**Texture Enhancement for Realistic Simulation** We further apply advanced image-processing techniques to enhance mesh textures. During mesh construction and simplification, textures from the original 3D Tiles are baked onto new geometries, which can result in missing or distorted regions. To address these issues, we first employ a Stable Diffusion 3 (Stability AI, 2024) based inpainting method to remove noise and repair damaged or incomplete textures. We then refine texture details using street-view imagery. This two-step process significantly improves visual fidelity, making textures more suitable for ground-level rendering.

**Object Replacement for Interactive Scene** To enhance scene interactivity, we combine generative and retrieval methods to populate the environment with interactive objects (e.g., bikes and tents). Using OSM annotations, we identify object types and locations to reflect real-world contexts. For relatively simple objects, such as tents, we adopt a generative pipeline that uses OSM text annotations on amenities as input: a Stable Diffusion model (Rombach et al., 2021) first generates images of the relevant objects, which are then processed by the One-2-3-45 framework (Liu et al., 2024) to produce corresponding 3D meshes. For more complex objects, such as trees, we use the retrieval pipeline, which randomly samples assets whose categories match the OSM annotations from a pre-collected dataset.

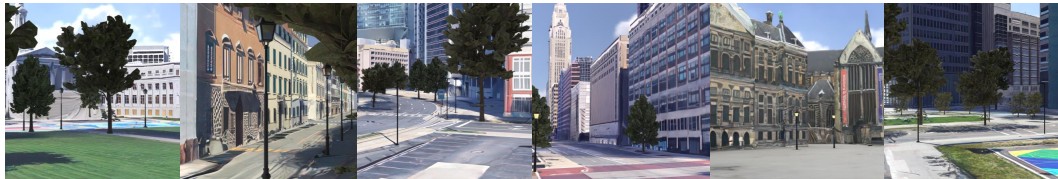

Figure 3: **Egocentric view of the generated scenes.** The resulting scene features clean geometry and realistic textures, which support physical simulation and enhance real-world style fidelity.

**Place and Transit Annotations with Geospatial Data** To facilitate alignment with real-world locations and provide semantic context for community activities, we developed a pipeline to automatically annotate places, buildings, and public transit within scenes using geospatial data. First, we query Google Maps Places for location information in the target area and organize results into different categories. Next, we use OSM to retrieve building names and bounding boxes, matching them with the place entries. We then filter out unmatched or inaccessible locations to generate accurate place annotations. Finally, we annotate bus transit routes based on these place annotations. These metadata enable agents to access location-specific information and support tasks that require spatial context, such as navigation and location-based decision-making, and also power traffic simulation, including buses, pedestrians, and other vehicles.

**Indoor Scenes Creation** To create indoor scenes in the communities, we employ a pipeline combining generation and retrieval to produce detailed, realistic multi-room environments. The pipeline's input is the building names in the target area, obtained from Google Maps and OSM. We first retrieve indoor layouts from GRUTopia (Wang et al., 2024a) for categories such as offices, restaurants, and stores. For building types not covered by GRUTopia, we use Architect (Wang et al., 2024b) to generate the corresponding indoor rooms for simulation.

### 3.2 AGENT COMMUNITY GENERATION

Given diverse generated scenes from real-world geospatial data, we introduce a generative pipeline to populate these environments with communities of agents endowed with grounded character profiles and social relationship networks, given their embodiments.

**Characters and Social Network Generation** We utilize the open-world knowledge of the Large Language Model (LLM) to generate agent character profiles and personalities grounded in the scene. Specifically, we use GPT-4o to perform this generation. The input to the LLM is structured into two parts to create characters grounded in a specific scene. The first part contains scene-related information, such as the scene name and details about various places, including their names, types, and functionalities. The second part includes details on the agents' appearances to ensure consistency between their visual attributes and generated profiles, which are annotated with the name and age. With both parts provided, the LLM generates agent profiles along with their social relationships. The profiles consist of basic attributes such as names, ages, occupations, personalities, and hobbies, which influence each agent's daily activities. Social relationships are structured as groups, each containing a subset of agents along with a text description and a designated place for group activities, connecting these agents into a cohesive community.

**Grounding Validator** We implemented a grounding validator that verifies whether generated character profiles are accurately grounded in the scene by ensuring all referenced places exist. If validation fails, the LLM receives feedback from the validator and attempts to correct the mismatch. Detailed examples of prompts used in the pipeline, generated characters, and social relationship networks are provided in the Appendix. K.

**Human-Like Avatar Creation** We first obtained 20 avatar skins representing diverse genders, professions, and appearances from Mixamo[*] for integration into the Virtual Community. We also used the Avatar SDK[*] to generate high-fidelity human meshes from synthetic human face images from FaceSynthetics (Wood et al., 2021), enabling representation of diverse individuals.

---

[*] https://www.mixamo.com/
[*] https://avatarsdk.com

### 3.3 Unified Simulation for Human and Robot Community

Virtual Community provides a unified framework for simulating both robots and human agents in the community. It implements an avatar simulation framework for human agents, while robot simulations are largely inherited from Genesis. Genesis is a universal physics engine for general-purpose embodied AI and robotics applications.

**Avatar Simulation and Control** To simulate avatars in Genesis physics engine, we combine SMPL-X human skeletons with these avatar skins to model human avatars. The motions of these avatars are parameterized by SMPL-X pose vectors $J \in \mathbb{R}^{162}$ and global translation and rotation vectors $T, R \in \mathbb{R}^3$. We use over 2,000 motion clips from Mixamo and adjust their playback speeds to match our avatars, including walking, object manipulation, and vehicle entry. For walking, we loop the clip until the avatar covers the required distance. For object-related actions, objects are kinematically attached to or detached from the avatars' hands based on the action. Similarly, during vehicle-related motions, avatars are kinematically attached to or detached from vehicles. We also incorporate physics constraints: collision detection is performed between avatars and scene entities, and motion terminates upon detection of a potential collision.

**Daily Schedule Generation and Simulation** Given the scene-grounded character profiles and social relationship networks, we prompt foundation models to generate each agent's daily schedule (Park et al., 2023). However, we structure each schedule so that every activity includes a start time, an end time, an activity description, and a corresponding location. We also explicitly account for the commute time between activities at different locations to reflect the actual cost of navigating an expansive 3D environment. This approach allows agents to follow the organized high-level plan effectively and maintain consistency over time. During simulation, agents follow the generated schedules to carry out daily activities. Examples of detailed daily plans are provided in the Appendix. B.2.

**Robot Agent Simulation** We simulate robots alongside avatars in the Genesis simulator. Virtual Community supports five types of robots: drones, quadruped robots, humanoid robots, wheeled robots, and mobile manipulators, each with a distinct robot controller. The robot controller bridges the interface between Virtual Community and Genesis, exposing only selected action spaces. Virtual Community shares the same simulation loop between avatars and robots with different control frequencies. To support faster collision detection during robot physics simulation, we use an invisible terrain mesh and decomposed building meshes as collision geometry for the background scene, enabling more efficient physics simulation.

## 4 Open World Multi-Agent Planning

Based on Virtual Community, we propose the *Community Planning Challenge* to evaluate multi-agent planning capabilities in outdoor and indoor environments. The challenge comprises three *community assistant* tasks, in which human agents cooperatively plan to assist multiple humans with daily open-world activities, and a *community influence task*, in which human agents competitively plan to efficiently connect and interact with other human agents in the community to increase their social influence.

### 4.1 Community Assistant Tasks

The community assistant tasks include the following three categories that require agents to plan cooperatively to provide humans with assistance on daily activities:

• **Carry:** Locate people and follow them to help carry objects to their home.

• **Delivery:** Move objects from source locations (indoor or outdoor) to a destination.

• **Search:** Locate target objects within an outdoor region or an indoor room.

**Task Settings** Each task includes multiple subtasks. For example, the Carry task requires agents to help humans carry several objects while following them. Therefore, adaptive task planning is essential for scheduling these runs, routing in the dynamic open world between waypoints, and managing task-level dependencies. We study two settings with different numbers of assistants: the *1-assistant*

Table 1: **Main results of the Multi-Agent Community Planning Challenge.** We report Success Rate (SR), Time Consumed (Ts), and Human Following Rate (HR) for three community assistance tasks averaged over 24 scenes.

| Method | Carry | | Delivery | | Search | | Avg SR↑ |
|---|---|---|---|---|---|---|---|
| | SR↑ | HR↑ | SR↑ | Ts↓ | SR↑ | Ts↓ | |
| 1-assistant | | | | | | | |
| Random | 0.0 | 0.0 | 0.0 | 1500.0 | 0.0 | 1500.0 | 0.0 |
| Heuristic | 34.7 | **16.5** | **46.5** | **1462.9** | 45.1 | 1440.3 | 42.1 |
| MCTS Planner | **42.3** | 7.7 | 39.6 | 1500.0 | 45.1 | 1500.0 | 42.4 |
| LLM Planner | 29.9 | 15.3 | 41.7 | 1500.0 | **70.1** | **1339.0** | **47.2** |
| 2-assistant | | | | | | | |
| Heuristic | **52.8** | **25.6** | **59.7** | **1415.8** | 51.4 | 1364.3 | **54.6** |
| MCTS Planner | 42.4 | 8.0 | 43.8 | 1500.0 | 48.6 | 1500.0 | 44.9 |
| LLM Planner | 30.2 | 10.7 | 43.8 | 1500.0 | **77.8** | **1141.3** | 50.6 |

setting, in which a single assistant needs to provide assistance to human agents in the community, and the *2-assistants* setting, in which two assistants plan cooperatively to provide assistance.

**Observation and Action Spaces** At each simulation step, agents are provided with an observation consisting of RGB-D images with the corresponding camera matrix, segmentations, current poses, and task information. The action space includes *move forward*, *turn left*, *turn right*, *enter/exit bus/bike*, and *communicate*. The movement and turning actions can be set with a continuous amount.

**Evaluation Metrics** We provide three evaluation metrics for all assistant tasks: success rate (SR), defined as the number of successful subtasks divided by the total number of subtasks; average time consumed (T) per task; and human following rate (HR) for the carry task, defined as the number of frames in which the agent follows a human within a specified distance range divided by the total number of frames during the task. When the simulation reaches a total step limit of 1500, it stops and the results are evaluated automatically.

**Baselines**

• **Perception and Navigation Module**: We implement all baseline agents within a hierarchical planner framework. The high-level actions include choosing subgoals, such as navigating to a specific person or building. Low-level actions include moving forward, turning, and object-related actions such as picking up items. All agents employ the same low-level navigation algorithm, which reconstructs a point cloud from egocentric RGB–D observations at each step and converts it into a volumetric grid representation at a resolution of $0.1\,\mathrm{m}$. Based on this grid, a 2D occupancy map with a resolution of $0.5\,\mathrm{m}$ is generated, and an A* algorithm is used to efficiently compute the shortest path to a bounding box. To accommodate dynamic environments, the navigation module recalculates the optimal path at every step.

• **Random Planner**: The Random planner is a trivial planner that randomly selects from the space of high-level actions without any planning.

• **Heuristic Planner**: The heuristic planner is based on a finite-state automaton defined by domain experts. At each state, the agent takes an action such as navigating, picking objects, entering rooms (see details in Appendix.F.4).

• **MCTS Planner**: We also introduce a new Monte-Carlo Tree Search (MCTS) based baseline planner, which employs Monte-Carlo Tree Search to optimize task plans.

• **LLM Planner**: We follow CoELA (Zhang et al., 2023) to design an LLM Planner with a modular framework driven by `gpt-4o` to generate and select subplans, including navigation to specified open-space locations, searching for objects, entering indoor areas, and performing object manipulation actions (pick and put). At each decision point, the LLM is prompted with the current state and task objectives and produces a subplan—a sequence of high-level actions (Song et al., 2022), which is then executed step by step by the agent. A communication module is also adopted to facilitate the cooperation among agents through natural language communication.

**Results** We evaluate the above baselines on all three tasks. The results in Table 1 demonstrate the challenges of planning in open-world environments. From our experiment, the *Random* baseline

Table 2: **Ablations on Distance Modeling (DM).** Without explicit distance modeling, the performance of both the LLM planner and the MCTS planner drops, with the decline being especially significant for the MCTS planner.

| Method | Carry | | Delivery | | Search | | Avg SR↑ |
|---|---|---|---|---|---|---|---|
| | SR↑ | HR↑ | SR↑ | Ts↓ | SR↑ | Ts↓ | |
| | | | 1-assistant | | | | |
| MCTS Planner w/o DM | 33.3 | 7.0 | 29.9 | 1500.0 | 23.9 | 1500.0 | 29.0 |
| LLM Planner w/o DM | 27.5 | 14.6 | 39.7 | 1500.0 | 66.0 | 1236.2 | 44.4 |
| | | | 2-assistant | | | | |
| MCTS Planner w/o DM | 34.0 | 6.9 | 27.1 | 1500.0 | 27.1 | 1500.0 | 29.4 |
| LLM Planner w/o DM | 25.0 | 15.2 | 55.6 | 1432.9 | 76.4 | 1060.2 | 52.3 |

fails to understand spatial relationships among tasks and no single method prevails for all the tasks. One common failure mode of baseline agents is underestimating the cost of open-world navigation and search, which leads to suboptimal task arrangement. Heuristic Planner performs strongly in the delivery task. While the LLM Agent prevails by a large margin in the search task, which involves no interaction with objects, it performs poorly in the other two tasks, where LLMs show difficulty tracking the task progress given only action history.

**Ablation study** To assess distance modeling, we ablate both the LLM planner and the MCTS planner, omitting the Heuristic planner since it lacks distance modeling. We remove spatial information from the LLM prompt and use a uniform distance heuristic for MCTS. As shown in Table 2, both degrade without distance modeling, with MCTS suffering far more. Notably, while the LLM planner drops in the 1-assistant setting, it can surpass its baseline in the 2-assistant setting, suggesting that cooperation helps offset missing distance modeling.

## 4.2 COMMUNITY INFLUENCE TASK

To further investigate agents' planning and social capabilities in open-world settings, we introduce the *Community Influence Task*—a novel, open-ended social challenge in which two main human agents compete to connect with and persuade other community members to form relationships with them. Due to differences in personality traits and social status, strategic planning is required to influence and shift member opinions over time.

**Task Settings** Each community contains two main agents and thirteen other members. The main agents must navigate the environment, locate potential members to connect with, and attempt to persuade them through dialogue. At the end of each day, every community member ranks their friendship level with the two main agents.

**Observation and Action Spaces** The observation and action spaces for all agents match those in the *Community Assistant Tasks*. Since this task focuses on social planning, both main agents are given access to the daily schedules of all members.

**Experimental Settings and Evaluation Metrics** We run experiments in five distinct communities. An agent is considered more effective at influencing others if it achieves a higher average friendship rank across all members.

**Baseline** We evaluate different LLMs as the planning backbone for the main agent. Given the daily schedules and character traits of community members, the main agent prompts the LLM to select the next member to visit, considering both spatial proximity and potential influence. After navigating to the target, the same LLM generates up to three rounds of conversation, conditioned on the main agent's and the target member's profiles. Once the conversation concludes, the main agent proceeds to the next selected target. Full prompt details are provided in the Appendix. K.

**Metrics** We use two metrics: (1) *Average friendship-ranking wins (Win.)* — the average win rate in the friendship ranking across all community members at the end of the day. Higher values indicate that an agent is more effective at forming new connections and expanding its social influence. (2) *Conversion rate (Conv.)* — the proportion of originally non-supporting members who become friends with the agent by the end of the day.

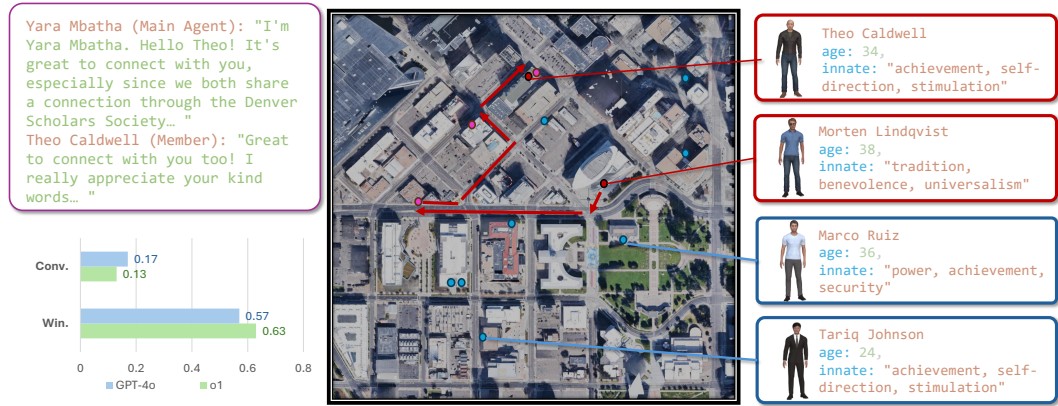

Figure 4: We evaluate baseline agents on the *Community Influence Task* across five communities. Results show that more powerful LLMs are better able to connect with and influence other members in the community.

**Results** As shown in Figure 4, the main agent with the `o1` backbone achieves higher average friendship rankings and conversion rates than the `gpt-4o` backbone [*], indicating greater ability to change members' opinions in most communities. For example, in the Denver community, the agent, Yara Mbatha, persuaded Theo by leveraging their shared affiliation with the Denver Scholars Society and emphasizing that common bond. We also observe that when starting with a large advantage, the `gpt-4o` agent sometimes fails to gain additional supporters, primarily due to suboptimal target selection and less effective persuasion strategies. These results suggest that, even with advanced LLMs, there remains substantial room to improve embodied agents' abilities in building social connections and exerting influence.

**Ablation study** To further understand the effects of the persuasion target selection and dialogue generation, we conduct an ablation study with two settings: (1) With the same backbone for target selection, use different backbones for dialogue generation. (2) With the same backbone for dialogue generation, use different backbones for selecting targets. The results in Table 3 show that both target selection and dialogue generation improve with a stronger backbone, while dialogue generation contributes more substantially to the overall performance gains.

Table 3: **Ablation experiments on the target-selection and dialogue components.**

| Dialogue | Target Selection | Win.↑ | Conv.↑ |
|----------|------------------|-------|--------|
| GPT-4o | GPT-4o | 0.57 | 0.17 |
| GPT-4o | o1 | 0.58 | 0.06 |
| o1 | GPT-4o | 0.60 | **0.20** |
| o1 | o1 | **0.63** | 0.13 |

## 5 OPEN WORLD MULTI-ROBOT COOPERATION

In addition to high-level planning and interactions, we also explore low-level physics challenges in multi-agent, open-world settings. In this section, we introduce the *Community Robot Challenge*, which features scenarios where two heterogeneous robots cooperate to assist humans in open-world environments.

**Task settings** The *Community Robot Challenge* builds upon the *Carry* and *Deliver* categories from the *Community Assistant* tasks (Section 4.1), introducing physics-level collaboration in open-world environments. In this challenge, robots must cooperate to deliver an object to a destination or assist a human avatar by picking up and carrying personal items while following the avatar.

**Robot Settings** We use two robot assistants: a mobile manipulator—based on the Google robot model in MuJoCo (Todorov et al., 2012), augmented with one degree of freedom for forward trans-

---

[*]We used GPT-4o-2024-11-20 and o1-2024-12-17 during experiments.

Table 4: **Detailed results of the Community Robot Challenge.** We report Success Rate (SR) and Time Consumed (Ts) for two community robot tasks averaged over 21 different scenes.

| Method | Carry | | Deliver | | Avg SR↑ |
|--------|-------|-------|---------|-------|---------|
| | SR↑ | Ts↓ | SR↑ | Ts↓ | |
| Heuristic | **17.6** | **126.9** | **22.2** | **129.4** | **19.9** |
| RL | 9.5 | 143.6 | 19.0 | 166.7 | 14.3 |
| Heuristic w Oracle Grasp | **23.5** | **124.4** | **50.0** | **131.2** | **36.8** |
| RL w Oracle Grasp | 19.0 | 149.7 | 42.9 | 168.1 | 31.0 |

lation and another for rotation about the z-axis—and a wheeled robot carrier with four degrees of freedom (one per wheel). In addition, Virtual Community supports quadruped and humanoid robots, which are described in the Appendix. D.

**Observation and Control Spaces** Observations include RGB–D images, segmentations, the base pose, and task-related information. The action space consists of 11 DoFs for the mobile manipulator (7 DoFs for the arm, 2 for the gripper, and 2 for locomotion) and 4 DoFs for the wheeled robot (1 for each wheel).

**Baseline Pipeline** We implement two baselines including *Heuristic* and *RL*. The heuristic baseline inherits the navigation module from baseline avatars in the Community Assistant Tasks 4.1. For navigation, robot computes a collision-free path with A* search. For manipulation, the robot solves for a feasible grasp pose with inverse kinematics, and plans and executes the grasp motion with RRT-Connect (Kuffner & LaValle, 2000). We also implemented a *VLA baseline*, which achieved near-zero performance. Full details are provided in the Appendix. F.6.

**Results** According to the results in Table 4, all baselines achieve higher scores on the delivery task than on the carry task, highlighting the added difficulty of simultaneously manipulating objects and following a human in a dynamic open-world environment. Moreover, without using an oracle grasp, performance drops significantly, underscoring the challenge posed by the manipulation component in this task. The reinforcement learning baseline performs worse than the heuristic baseline, which uses inverse kinematics and RRTConnect to compute manipulation trajectories. This gap is because the classical planner explicitly solves for optimal paths in configuration space, whereas the RL agent must discover effective control sequences under sparse reward signals.

**Ablation study** We additionally evaluate a decomposed RL variant that trains separate reach and place policies. As shown in Table 5, decomposition slightly improves the performance.

Table 5: **Ablation experiments on the RL baseline.** Decomposing task with different policies leads to minor improvements on both tasks.

| Method | Carry | | Deliver | | Avg SR↑ |
|--------|-------|-------|---------|-------|---------|
| | SR↑ | Ts↓ | SR↑ | Ts↓ | |
| RL w/ Oracle Grasp | **19.0** | 149.7 | 42.9 | **168.1** | 31.0 |
| RL Decomposed w/ Oracle Grasp | **19.0** | **149.0** | **47.6** | 172.1 | **33.3** |

## 6 CONCLUSION

We introduce Virtual Community, an open-world simulation platform for multi-agent embodied AI that supports scalable, simulation-ready generation of open-world scenes and agent communities, along with physically realistic simulation of multiple embodied avatars and robots. As an initial demonstration, we propose two novel open-world multi-agent challenges—the **Community Planning Challenge** and the **Community Robot Challenge**—each developed and evaluated using a variety of baseline methods. One limitation of this work is that the outdoor scenes are not modeled in sufficient detail to accurately reflect the physical and visual properties of real-world environments. We hope Virtual Community will advance embodied AI research toward embodied general intelligence capable of handling real-world complexities and coexisting with human communities.

## ACKNOWLEDGEMENT

This project is supported by MURI N000142412748, NSF IIS-2441250, and NSF IIS-2404386. We gratefully acknowledge Haoyu Zhen, Ruxi Deng, Hao Zou, Tiger Harburg, and Siyuan Cen for their efforts in the project website and data collection. We thank Chunru Lin, Yian Wang, Xiaowen Qiu for their insightful discussion and Zhou Xian, Yuncong Yang, Zeyuan Yang, Jiaben Chen for their feedback on the project.

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

# Appendix

## A  3D SCENE GENERATION DETAILS

In this section, we present the implementation details of the 3D scene generation pipeline. This pipeline takes three inputs—latitude, longitude, and range radius—to generate 3D community scenes for simulation, automatically outputting multiple textured 3D meshes, including buildings, terrain, roofs, and a JSON format of objects that can be loaded in the physics engine. The entire scene generation pipeline is implemented in Blender[*].

### A.1  MESH PREPROCESSING AND TERRAIN CONSTRUCTION

Given the latitude, longitude, and range radius, we first retrieve the 3D tiles data within the specified range. The original 3D tiles data is in the Earth-Centered Earth-Fixed (ECEF) coordinate system, which is then converted to East-North-Up (ENU) coordinates, setting the mesh centroid as the origin by averaging the positions of all vertices. After this translation, we seamlessly join all tile meshes by merging each vertex near the boundary of one tile with the nearest vertex of an adjacent tile. With these preprocessing steps, we obtain a coordinate-aligned and integrated mesh for each scene.

To construct the corresponding terrain mesh, we retrieve OpenStreetMap (OSM) road and ground annotations within the specified area and align them with the mesh obtained from the previous step. By sampling latitude-longitude pairs along roads and ground surfaces and performing ray casting from these sampled points, we calculate the heightfield for each road and ground area. However, the heightfield can be noisy due to extraneous meshes, such as cars or other objects, in the 3D tiles data. To address this, we apply a rule-based filter to remove abnormal points from the heightfield. The terrain mesh is then constructed using bilinear interpolation on the cleaned heightfield.

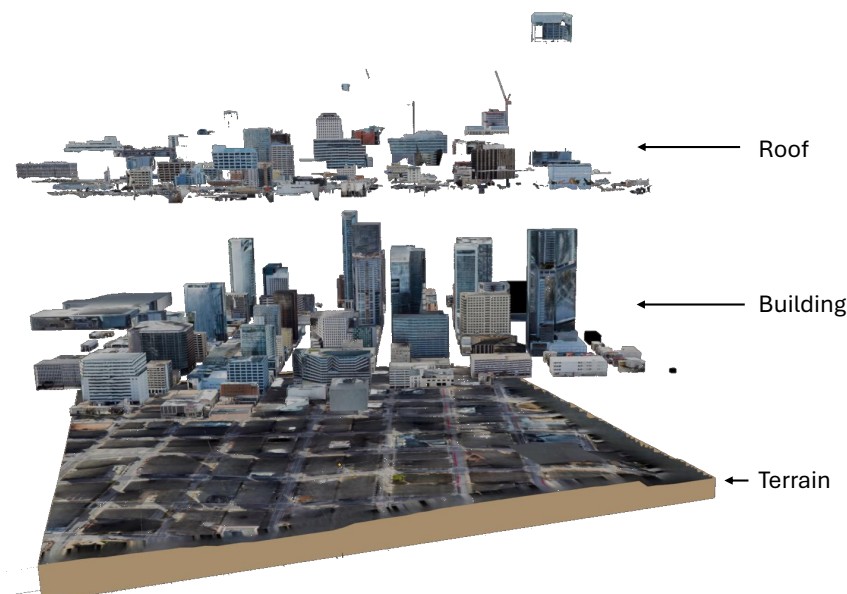

Figure 5: We decompose the outdoor 3D scene into three components - the terrain, buildings, and roofs. Each part is generated separately using different strategies to balance the visual appearance and geometry complexity.

---

[*] https://blender.org/

## A.2 TEXTURE TRANSFER AND ENHANCEMENT

The meshes in the 3D tiles data have sub-optimal geometry and noisy textures for simulation since they are reconstructed by photometric methods from aerial photos. To generate meshes suitable for simulators, we apply the *texture transfer* and *texture enhancement* step to the 3D tiles data.

**Texture Transfer** To create topologically sound geometry for the simulator, we first construct geometries called *OSM geoms* using the *Simple 3D Buildings*[*] annotation from OpenStreetMap overpass API. These OSM geoms have significantly fewer vertices and faces compared to the 3D tiles while ensuring water-tightness and topological soundness. However, these geometries constructed by the rule-based method do not contain any texture information. We address this deficiency by baking the texture from 3D tiles to the OSM geoms using Blender. Specifically, we used the caged baking method to improve the texture transfer quality further.

**Texture Enahancement** Since the original texture does not have sufficient resolution for photo-realistic first-person-view rendering, we apply diffusion models such as StableDiffusion for inpainting and super-resolution (Rombach et al., 2021).

a)   b)

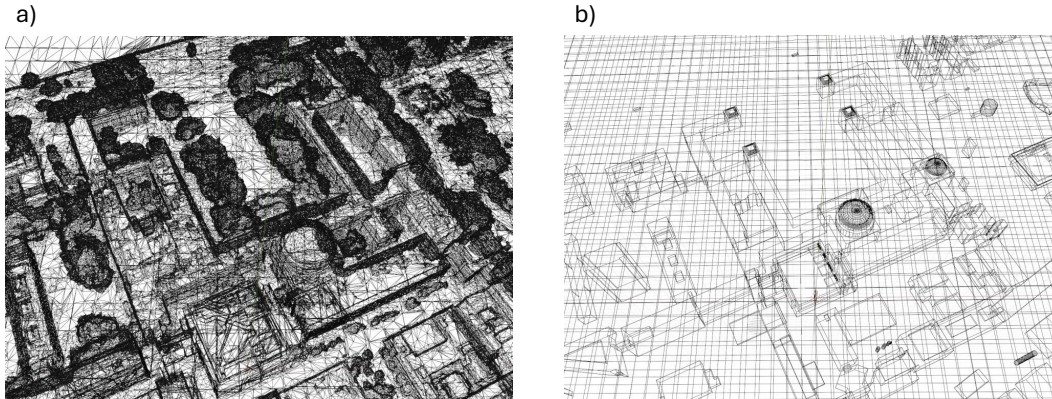

Figure 6: Comparing the scene geometry of a) raw Google 3D Tiles and b) our reconstructed scene at the Boston, MA. Our scene has much simpler geometry and reliable mesh topology, facilitating physical simulation at scale.

## A.3 STREET VIEW REPROJECTION DETAILS

To further enhance the realism of building and terrain texture, we utilize the Street Views from Google StreetView[*] and Mapillary[*] to fine-tune the scene texture. This process, called Street View reprojection, is composed of four major steps: 1) camera initiation, 2) view fine-tuning, 3) street-view inpainting, and 4) texture reprojection.

**Camera Initilization** In this step, we fetch all the street-view images in the range of 3D scenes. Using the provided metadata on longitude, latitude, orientation, and camera configuration, we instantiate cameras with corresponding intrinsic and extrinsic matrices in Blender.

**View Fine-Tuning** Since the image metadata are prone to sensor measurement noise, we perform an additional step of view fine-tuning to perform minor pose correction on the camera in Blender. For each camera placed in the initialization step, we first render the 3D scene mesh from the camera's perspective. The rendered image is then matched with the street-view image using LoFTR matching algorithm (Sun et al., 2021). Using the matched keypoint pair, we are able to solve the pose difference between the street-view image and the rendering camera following the 5-point method (Nistér, 2004). The pose correction estimated by the 5-point method is then validated by comparing the norm of $\mathfrak{se}(3)$ pose vector with a pre-defined threshold.

---

[*]https://wiki.openstreetmap.org/wiki/Simple_3D_Buildings
[*]https://www.google.com/streetview/
[*]https://www.mapillary.com

Table 6: Average counts of object categories per scene.

| Object category | Average count |
|---|---|
| Amenities objects | 133 |
| Natural objects | 357 |
| Transit objects | 9 |

Table 7: Averaged number of annotated elements per scene.

| Annotated element | Avg. number per scene |
|---|---|
| Buildings | 53 |
| Places | 85 |
| Roads | 473 |
| Junctions | 76 |
| Restaurant | 25 |
| Store | 21 |
| Accommodation | 16 |
| Transit | 9 |
| Office | 8 |
| Entertainment | 5 |
| Open | 1 |

**Streetview Inpainting** The street-view images often contain dynamic objects such as vehicles and pedestrians. These objects are intrinsically dynamic and do not have corresponding geometry on the reconstructed 3D scene. Therefore, we identify these objects using Language grounded Segment Anything (LangSAM) (Ren et al., 2024) and perform inpainting using StableDiffusion.

**Texture Reprojection** Finally, we project the inpainted street-view images to the 3D meshes using the corrected camera pose and *Texture Painting* feature from Blender. Since the street-view images are taken under various lighting and weather conditions, the color distribution between images and 3D scene texture may differ significantly. To mitigate the gap between color distributions, we implement a color-correction routine that automatically aligns the color temperature and exposure of multiple images using histogram alignment. We also apply performance optimizations, including view-frustum clipping and greedy camera-mesh matching to speed up this process.

## A.4 INDOOR SCENE GENERATION PIPELINE

The Indoor Scene Generation Pipeline comprises two main stages to create detailed, realistic multi-room environments. Given building names in the target area—fetched from Google Maps and OpenStreetMap—as input, it outputs corresponding indoor scenes loadable in the simulator. In the retrieval stage, we query GRUTopia (Wang et al., 2024a) for the most relevant indoor layout, but because GRUTopia scenes can be extremely complex, we employ the generative stage for all but the most frequently used scenes. In the generative stage, following (Wang et al., 2024b), a diffusion-based inpainting model populates empty rooms with large objects, which are then detected and spatially positioned by vision models; subsequently, large-language models assist in selecting and placing suitable small objects onto or within the arranged large objects.

## A.5 QUANTITATIVE STATISTICS OF GENERATED SCENES

Currently, Virtual Community provides 33 amenity types and common city objects, along with two special transit amenities, including bus stop and bicycle station. Table 6 shows the statistics of object categories. The generated scenes are automatically annotated with places, buildings, and public transit within scenes using map data. Table 7 shows the statistics of annotated elements per scene.

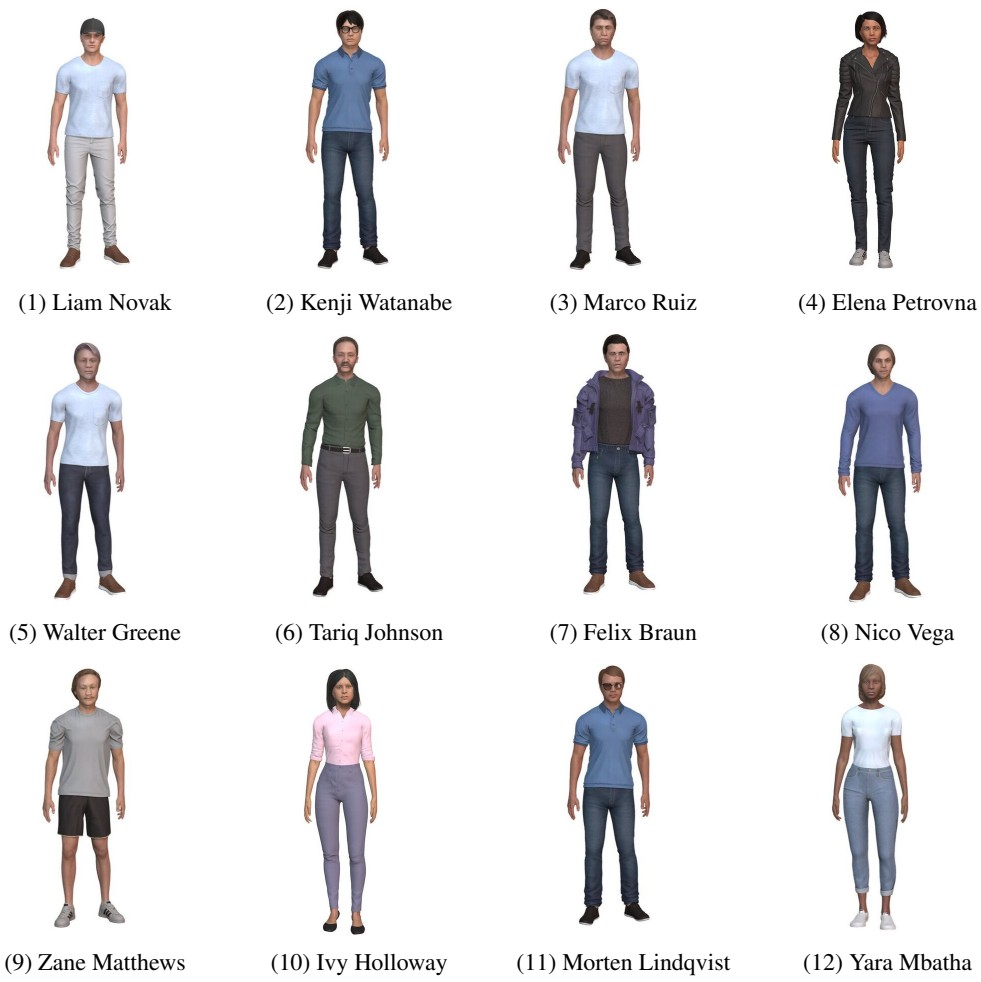

|                    |                      |                 |                      |
|--------------------|----------------------|-----------------|----------------------|
| (1) Liam Novak     | (2) Kenji Watanabe   | (3) Marco Ruiz  | (4) Elena Petrovna   |
| (5) Walter Greene  | (6) Tariq Johnson    | (7) Felix Braun | (8) Nico Vega        |
| (9) Zane Matthews  | (10) Ivy Holloway    | (11) Morten Lindqvist | (12) Yara Mbatha |

Figure 7: Some examples of generated avatars. These names have been randomly generated and do not correspond to any real individuals.

## B    AVATAR SIMULATION

### B.1    AVATAR MODELS AND MOTIONS

To simulate avatars in Virtual Community, we download FBX files from Mixamo that record human skeletal motion sequences and parse them into a hierarchical structure of human joints. Each skeleton joint is mapped one by one with the joints of the SMPL-X model. Then, we recursively traverse the joint tree structure to calculate the global coordinate system vector for each joint after its rotation at each time step $t$, and use this to drive the movement of the human skeleton. Based on these pose representations, each avatar's skin mesh is computed via forward kinematics.

We present examples of generated humanoid avatars in Figure 7. These demonstrates the capability of our method to create detailed and varied human-like avatars. Each skin model of characters includes 71 skeletal joints and can be adapted to animation sequences in SMPL-X and FBX formats. To reduce the computational load during animation playback in the Virtual Community, we optimized the skin models by applying Blender's Decimate Modifier tool, reducing the number of vertices in the 3D skin mesh by 90%.

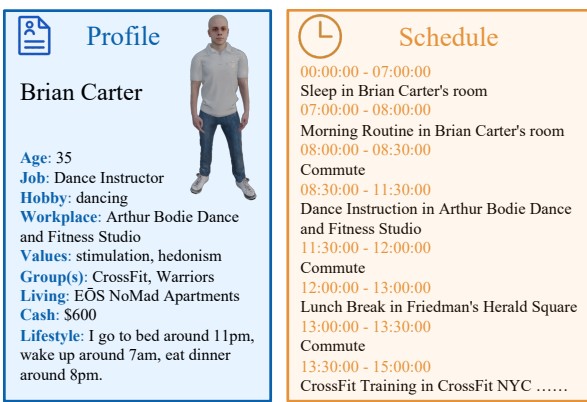

Figure 8: An example of generated character and daily schedule.

## B.2 PROFILE AND DAILY PLAN GENERATION

An example character with social relationship networks and daily schedule generated is shown in Figure 8 (a). Given the scene-grounded characters and social relationship networks, we prompt the foundation models to generate the daily schedule for each agent, using a similar design to (Park et al., 2023). Differently, we generate the daily schedule in a structured manner directly with each activity represented with a start time, an ending time, an activity description, and the corresponding activity place, and consider the required commute time between adjacent activities that are happening in different places explicitly, due to the actual cost of navigating in an expansive 3D environment. During character initialization, we use a flood-fill-like algorithm to verify every candidate starting position and reject any point that is not connected to the map boundary. This guarantees global reachability for all initialized agents.

## C    TRAFFIC SIMULATION

In this section, we present a detailed description of our traffic simulation implementation. The simulation pipeline includes two components: road network construction and traffic control. Together, these modules enable realistic and efficient urban traffic simulation.

### C.1    ROAD NETWORK

To implement traffic simulation, the first step is to construct an accurate and structured representation of the urban road network. Our system uses data obtained from the OSM API. Based on the raw OSM data, we build a comprehensive road information database that includes attributes such as road type, location, and width for each segment of the network.

For more advanced traffic simulation and control, we further convert the raw OSM data into the OpenDRIVE format. This format provides a highly structured and semantic-rich description of the road network, including road direction, geometry, lane configurations, and connectivity between roads. These features are essential for enabling precise vehicle navigation and traffic behavior modeling. Specifically, we use the OSM to OpenDrive converter provided by CARLA (Dosovitskiy et al., 2017).

### C.2    TRAFFIC CONTROL

Once we have the road network, pedestrians and vehicles can be placed either manually or randomly on the map. To make their behaviors realistic and coherent, we implement a Traffic Manager module responsible for controlling and coordinating all traffic participants.

The main functions of the Traffic Manager include path planning, collision avoidance, and traffic flow regulation. It ensures that both vehicles and pedestrians follow reasonable movement patterns

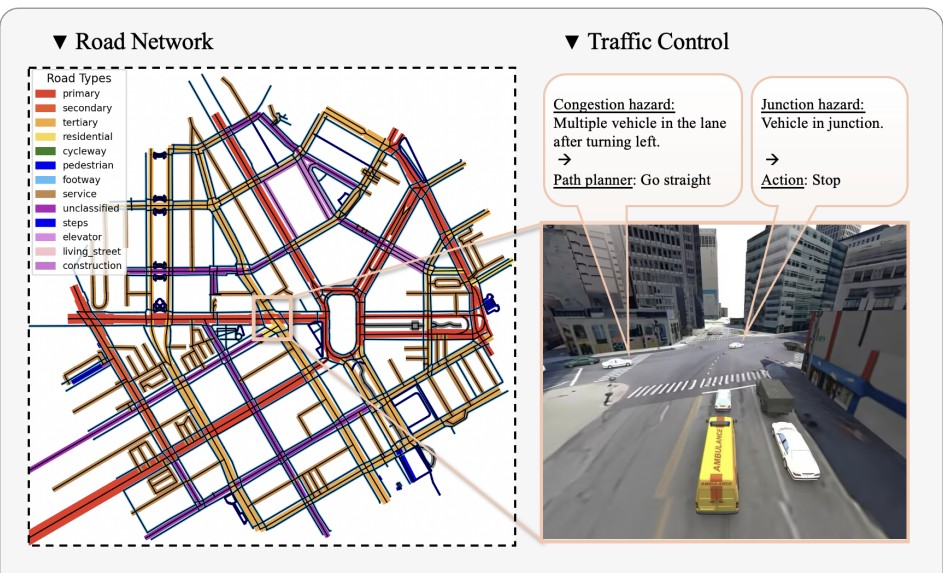

Figure 9: An example of the road map (left) and the corresponding traffic flow at one of its junctions (right).

while maintaining safety and efficiency. Considering both realism and computational performance, we define a set of simplified traffic rules within the Traffic Manager:

- **Junction Access Control** If any pedestrian or vehicle is currently inside an junction, no other agent will enter until the junction is clear.
- **Direction Preference** When a vehicle reaches an intersection and has multiple directions to choose from, it selects the route with fewer vehicles to minimize congestion.
- **Pedestrian Behavior** Pedestrians are allowed to walk in both directions along sidewalks. When two pedestrians come too close, they adjust their positions to avoid collisions.
- **Lane Change under Congestion** In cases where a lane becomes congested, some vehicles are allowed to switch to adjacent lanes to maintain traffic flow.

Figure 9 shows a road map and part of the traffic flow in Detroit. Since the traffic simulation takes OSM data as input, it can be generated in any scene where OSM data is available.

## D    ROBOT SIMULATION

In the Community Robot Challenge, we employ a robot carrier and a mobile manipulator. Although Virtual Community also supports other robot types—including quadruped and humanoid robots—and can readily accommodate any robot platform thanks to its Genesis foundation, in this section we describe the simulation details for the four default robot types.

### D.1    MOBILE MANIPULATOR

We adopt the Google Robot from the MuJoCo library and integrate it into Genesis as the default mobile manipulator. Following AI2-THOR (Kolve et al., 2017), Habitat (Savva et al., 2019), and ManiSkill3 (Tao et al., 2024), we add one joint to control forward/backward motion and another joint to enable rotation about the z-axis at the base.

### D.2    QUADRUPED AND HUMANOID ROBOTS

The Unitree Go2 serves as our default quadruped, and the Unitree H1 as our default humanoid, shown in Figure 10. We utilize the corresponding URDF files supported by Genesis (Authors,

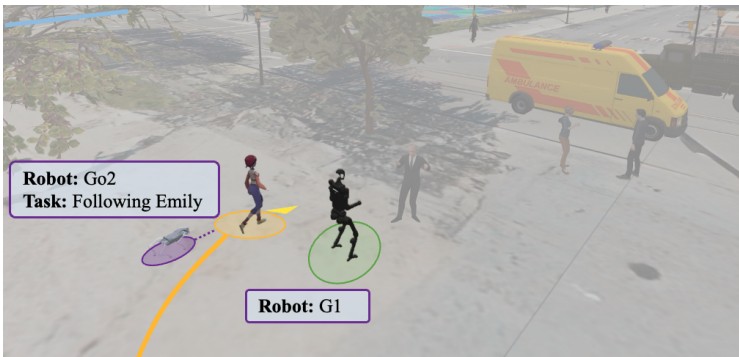

Figure 10: An example of simulating quadruped and humanoid robots in Virtual Community.

2024), together with reinforcement-learning-trained locomotion policies provided by the Genesis team.

### D.3 ROBOT CARRIER

We use the Husky robot as the default carrier, importing its URDF file from Bullet[*]. The carrier has four degrees of freedom—one per wheel. We have modified its top surface so that any object landing on it is automatically attached to the carrier.

### D.4 NESTED LOOP FOR ROBOT SIMULATION

To simultaneously support simulation of avatars with lower control frequencies and robots with higher control frequencies, we employ a nested framework in the simulation loop. The outer loop, which corresponds to one second of simulation time, handles the observation–action cycle for avatars. Each outer loop consists of 100 inner-loop steps for robots; in each inner step, we execute one physics frame in Genesis, one avatar-animation frame, and one observation–control step for the robots.

## E BENCHMARKING

### E.1 SCENE BENCHMARKING

In this section, we provide both quantitative and qualitative summaries of the scenes generated for the Virtual Community. Currently, all outdoor 3D scenes have a size of 800m × 800m. We generated 35 diverse scenes covering 17 countries from North America, Europe, and Oceania. We show some of the generation results in Figure 11 and Figure 12.

To assess the quantitative quality of our generated scenes, we compare them with the original Google 3D Tiles data along two dimensions: visual fidelity and geometric complexity. Visual fidelity directly impacts the ego-view observations received by agents, whereas geometric complexity affects the difficulty of physics simulations. For visual fidelity, we compute the Fréchet Inception Distance (**FID**) (Heusel et al., 2017) and Kernel Inception Distance (**KID**) (Bińkowski et al., 2018) between our generated scenes (and the baseline scenes) and Google Street View images captured at identical camera positions and orientations. For geometric complexity, we measure the average number of mesh faces per scene—excluding roof faces, which are not involved in the physics simulation—and compare these values between our scenes and the baseline. We rendered 31k images per method to evaluate visual fidelity, and utilize the 3D meshes of all 35 generated scenes to measure geometric complexity. According to the results in Table 8, our generated scenes has significantly improved the visual effects and reduced the geometric complexity compared with the origianl data.

---

[*]https://github.com/bulletphysics/bullet3

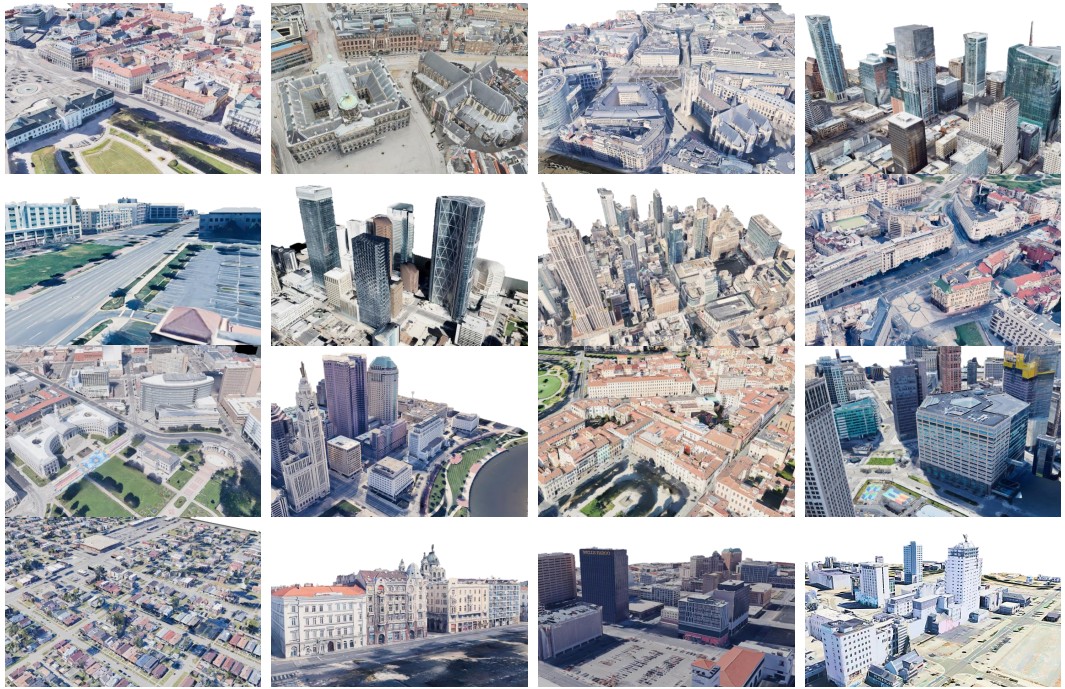

Figure 11: Images rendered from the generated large-scale scenes in North America, Europe, and Oceania. For each location, we generated high-quality scenes with an area of 640,000 $m^2$.

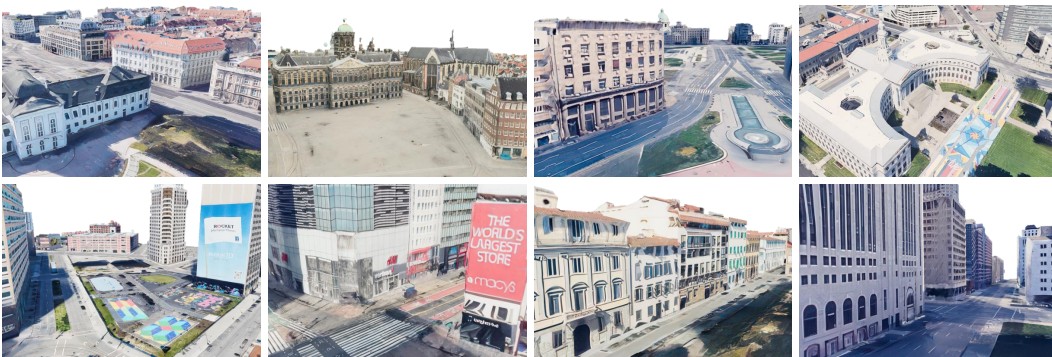

Figure 12: Close-up view of the generated scenes. The resulting scene has clean geometry and realistic texture, which is essential for physical simulation.

Table 8: We evaluate the generated scenes using Fréchet Inception Distance (FID) and Kernel Inception Distance (KID) for visual fidelity, and the average mesh face count for geometric complexity.

| Methods | FID↓ | KID ($\times 10^{-2}$)↓ | Face Num. ($\times 10^5$)↓ |
|---|---|---|---|
| 3D Tiles | 108.04 | 8.88 ± 0.66 | 20.94 |
| Ours | **83.65** | **7.60 ± 0.63** | **3.76** |

### E.2 SIMULATION BENCHMARKING

**Speed benchmarking** We benchmark the simulation speed of Virtual Community with the following settings:

- **RGB Setting**: The simulator provides avatar observations, including RGB signals, at 1 Hz, which encompasses 100 physics frames per second. We record the average physics frames per second (FPS) in this setting.

- **Depth Setting**: Similar to Habitat 3.0 (Puig et al., 2023), we adopt a depth-only configuration that renders a depth image at each physics step (at 100 Hz). In this setting, we also record the average physics frames per second (FPS).

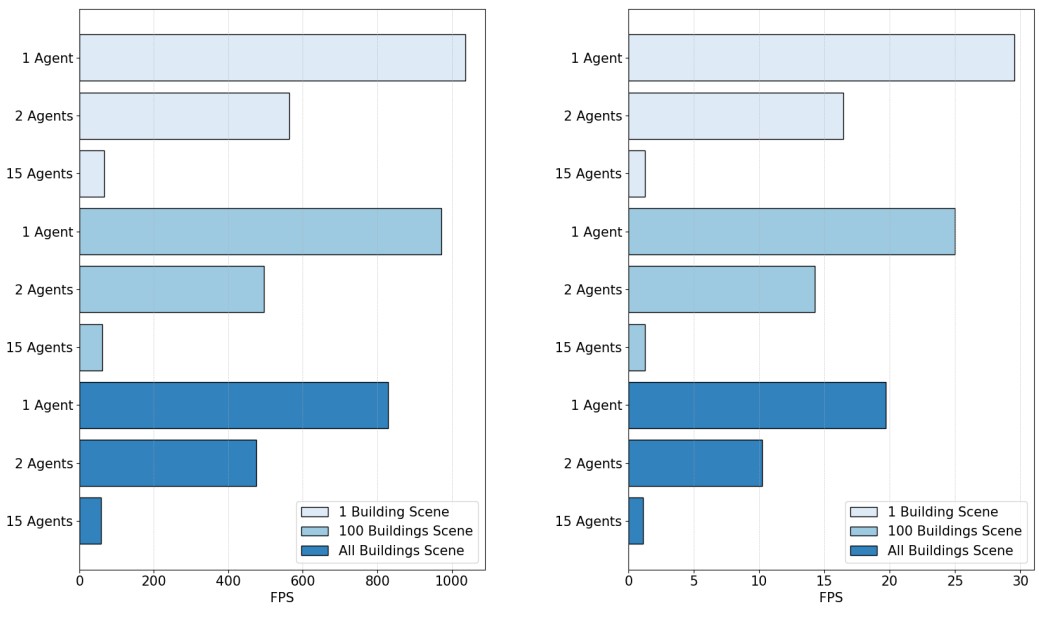

(a) RGB Setting Benchmarking Results      (b) Depth Setting Benchmarking Results

Figure 13: Simulation Speed Benchmarking under RGB and Depth Settings

We ran all experiments using a single process on an NVIDIA A100 GPU. Figure 13 presents the benchmarking results under various conditions. In each experiment, we load a fixed terrain background while varying the number of buildings. We also evaluate simulation speed as a function of the number of simulated avatar agents.

**Memory benchmarking** We measure the memory footprint of Virtual Community as a function of the number of simulated avatar agents while keeping the simulator configuration fixed. In each run, we load the full scene assets and record *peak* and *average* RAM usage and *peak* GPU memory usage during execution. All experiments are conducted with a single process on an NVIDIA L40S GPU. The results are summarized in Table 14.

Overall, memory usage scales approximately linearly with the number of agents. When increasing from 1 to 50 agents, peak RAM grows from 17.45 GB to 38.46 GB, average RAM grows from 15.11 GB to 36.01 GB, and peak GPU memory grows from 4.38 GB to 14.26 GB. This corresponds to an average increase of roughly 0.43 GB peak RAM, 0.43 GB average RAM, and 0.20 GB peak GPU memory per additional agent, indicating stable and predictable memory scaling for large agent communities.

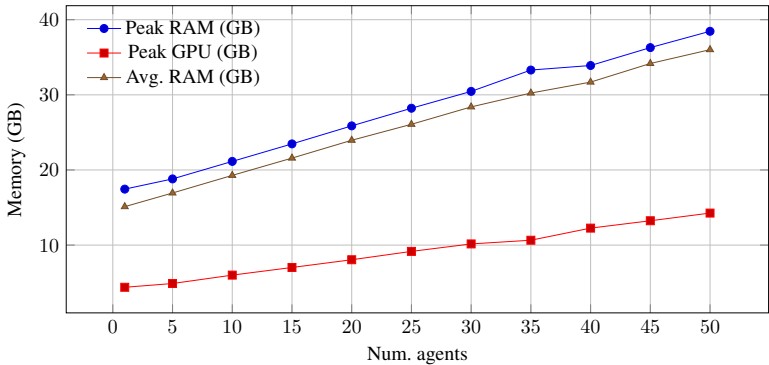

Figure 14: Memory benchmarking of Virtual Community.

## F  CHALLENGE DETAILS

### F.1  AVATAR OBSERVATION SPACE

Virtual Community provides each agent with the following observations at each frame:

- **RGB**: Visual input with dimensions of 256×256 and 3 channels.
- **Depth**: Depth information represented as a 256×256 single-channel map.
- **Segmentation**: Segmentation information represented as a 256×256 single-channel map.
- **Pose**: A 6D vector containing the 3D location and a 3D facing vector in ENU coordinates.
- **Camera Extrinsics**: Parameters defining the camera's position and orientation.
- **Events**: Text messages sent from nearby agents using the *communicate* action.
- **Other States**: Includes current cash, accessible locations, and action status.

### F.2  TASK GENERATION

For both assistant tasks in the *Community Planning Challenge* and the *Community Robot Challenge*, we employ a procedural task-generation pipeline that produces tasks for all scenes. The pipeline consists of:

- **Place Selection**: When a task requires a specific location (e.g., the destination in a carry task), we use the agent's profile and a list of known places as inputs, and prompt a large language model to select a valid target. Outdoor regions are included among the options.
- **Object Selection**: To determine the target object, we prompt the language model with the task description and provide nine candidate object types for it to choose from.
- **Object Placement**: After the object type and place are chosen, we retrieve the corresponding asset and position it appropriately.
  - *Outdoor locations*: placed at a random point not occluded by any building.
  - *Indoor locations*: placed on the surface of a randomly selected semantic container (floor, sofa, table, chair, desk, or bed).
- **Evaluation**: Once the object is placed, the pipeline automatically generates evaluation metadata (e.g., bounding boxes for the target object or agent). After the agent signals task completion, the simulator checks whether the success criteria are met and records the result.

### F.3  ADVANCED SETTING OF THE COMMUNITY PLANNING CHALLENGE

To provide comprehensive benchmarking in the *Community Planning Challenge*, we also introduce an advanced setting by increasing the source regions and restricting the distance-to-target constraint.

Table 9: **Results of the advanced Community Planning Challenge.** We report Success Rate (SR), Time Consumed (Ts), and Human Following Rate (HR) for three community assistance tasks averaged over three scenes.

| Method | Carry | | Delivery | | Search | | Avg SR↑ |
|---|---|---|---|---|---|---|---|
| | SR↑ | HR↑ | SR↑ | Ts↓ | SR↑ | Ts↓ | |
| Random | 0.00 | 0.00 | 0.00 | 1500 | 0.00 | 1500 | 0.00 |
| Heuristic | 0.00 | 0.00 | 16.7 | 1500 | 0.00 | 1500 | 5.55 |
| LLM Planner | 0.00 | 0.00 | 16.7 | 1500 | 27.7 | 1493 | 14.8 |
| HP + HP | **5.55** | 0.00 | 16.7 | 1500 | 5.55 | 1500 | 9.26 |
| LLM + LLM | 0.00 | 0.6 | 11.1 | 1500 | **38.8** | **1368** | 16.7 |
| HP + LLM | **5.55** | **1.6** | **22.2** | 1500 | 27.8 | 1412 | **18.5** |

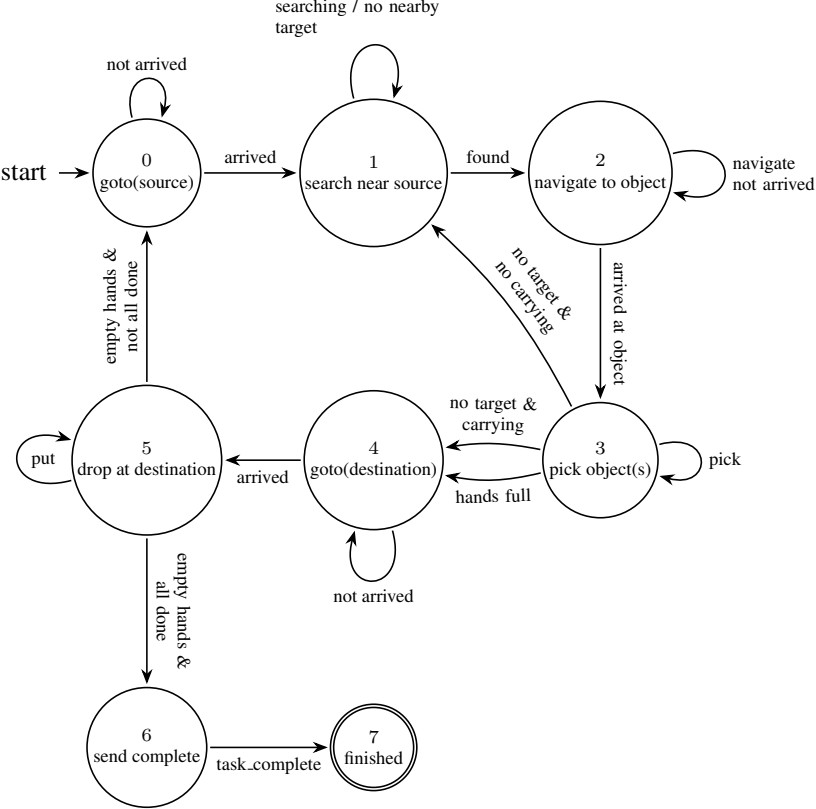

Figure 15: Finite-state automaton of the heuristic planner logic on the delivery task.

Table 9 shows the results for the simplified setting. The performance of baseline methods drops significantly in the advanced setting, indicating the increased difficulty in a larger task region. We also evaluate different combinations in the 2-assistant setting, where *HP+LLM* outperforms all other baselines, while *LLM+LLM* fails to generate an optimal plan in open-world scenes, highlighting the challenges of cooperative planning in the *Community Planning Challenge*. Most baselines struggle to complete tasks within the step limit, resulting in time consumption that approaches this maximum.

## F.4 IMPLEMENTATION OF THE HEURISTIC PLANNER

The heuristic agents use a finite-state automaton defined by human experts for each task. For example, Figure 15 shows the execution logic for the deliver task. With high-level actions such as

searching parsed as functions, we implement different automata for all three tasks with different logic.

## F.5 REWARD DESIGN FOR COMMUNITY ROBOT CHALLENGE

**Reward Function.** We design a shaped reward for the pick-and-place task of the mobile manipulator. The reward integrates distance penalties, grasp bonuses, and placement criteria:

- **Distance penalty to object.** At each step, the agent is penalized by the clamped Euclidean distance between gripper and object:

$$r_{\text{obj}} = -\min(\|p_{\text{gripper}} - p_{\text{object}}\|, 10.0) \times 0.02.$$

- **Distance penalty to goal (optional).** A similar penalty can be added for the object–goal distance:

$$r_{\text{goal}} = -\min(\|p_{\text{object}} - p_{\text{goal}}\|, 10.0) \times 0.02.$$

- **Grasp success bonus.** When the object is lifted above $2\times$ its size and the gripper is closed, a binary grasp reward is given:

$$r_{\text{grasp}} = \mathbb{1}\{\text{cube lifted and gripper closed}\} \times 5.0.$$

- **Placement-hold bonus.** Once the cube stays within $0.1\,\text{m}$ of the goal for 100 consecutive steps, a one-time placement reward is triggered:

$$r_{\text{place}} = \mathbb{1}\{\text{cube held at goal for 100 steps}\} \times 100.0.$$

The total reward is the sum of the active components:

$$r_t = r_{\text{obj}} + r_{\text{goal}} + r_{\text{grasp}} + r_{\text{place}}.$$

**Training Details.** We use Proximal Policy Optimization (PPO) implemented in rsl_rl (Schwarke et al., 2025). The main settings are summarized in Table 10.

Table 10: RL training configuration for the community robot task.

| Aspect | Configuration |
|---|---|
| **Observation space** | Joint pos/vel, robot pos, object pos, relative vectors ($\approx 2\times$DoF + 15 dims) |
| **Action space** | Continuous joint increments (all DoFs except base) |
| **Policy network** | Actor–critic, hidden dims [256, 256, 128], ELU activation |
| **Algorithm** | PPO, clip 0.2, entropy coef 0.001, value loss coef 1.0 |
| **Discount / GAE** | $\gamma = 0.99$, $\lambda = 0.95$ |
| **Learning rate** | $3 \times 10^{-4}$ (adaptive schedule) |
| **Batching** | 24 steps/env, 1024 envs, 5 epochs/update, 4 minibatches |
| **Episode length** | 250 steps max |
| **Iterations** | 500 |

## F.6 VLA BASELINE IMPLEMENTATION

We also implemented a *Vision-Language-Action (VLA) baseline* for the mobile manipulator in the Community Robot Challenge. This baseline is built upon the $\pi_0$ model (Black et al., 2024a), which provides a pretrained backbone model for physical manipulation tasks. We finetuned the model using a dataset collected in Virtual Community, where the initial robot poses were randomly sampled at the beginning of each episode to ensure diversity of starting conditions. In total, we collected approximately 3,000 successful trajectories of pick-and-place subtasks. Following the original implementation in $\pi_0$, We applied filtering to remove datapoints with actions which maximum dimension smaller than 0.01, selected success-only trajectories, and clipped values between the $p_{0.01}$ and $p_{0.99}$ percentiles.

Table 11: **Detailed results of the Community Robot Challenge, including the VLA baseline.** We report Success Rate (SR) and Time Consumed (Ts) for two community robot tasks averaged over 21 different scenes.

| Method | Carry | | Deliver | | Avg SR↑ |
|---|---|---|---|---|---|
| | SR↑ | Ts↓ | SR↑ | Ts↓ | |
| Heuristic | 17.6 | 126.9 | 22.2 | **129.4** | 19.9 |
| Heuristic w Oracle Grasp | **23.5** | **124.4** | **50.0** | 131.2 | **36.8** |
| RL w Oracle Grasp | 19.0 | 149.7 | 42.9 | 168.1 | 31.0 |
| VLA w Oracle Grasp | 0.0 | 1500.0 | 4.8 | 1434.4 | 2.4 |

**Input and Output Spaces.** The VLA model receives multimodal observations at each step, including:

- RGB images rendered at $224 \times 224$ resolution,
- Joint positions and velocities of the manipulator.

The model outputs joint increments $\Delta q$ for all controllable degrees of freedom. Actions are executed at a control frequency of 20 Hz.

**Training Setup.** We finetuned the model on collected trajectories using supervised behavior cloning. We used a learning rate of $5e-5$ and a batch size of 32. Training was performed for $5e4$ steps with early stopping based on validation loss.

**Performance.** As shown in Table 11, despite successful finetuning, the VLA baseline achieves near-zero performance in the Community Robot Challenge tasks. We identify three main reasons: (1) our tasks involve picking objects placed on the ground, whereas the pretrained VLA model is primarily trained on tabletop manipulation, leading to a large task gap; (2) our environments are outdoor scenes with small target objects, creating a significant visual domain gap; (3) the finetuning dataset size and training iterations are insufficient compared to full from-scratch training, and therefore cannot bridge the gap caused by the domain shift.

# G SINGLE AGENT TASK

We introduce *Community Commute*, a single-agent task, to illustrate open-world city environments, traffic systems, and the execution of an agent's daily plan in Virtual Community. This single-agent task focuses on daily-life simulation for an individual agent. While providing this single-agent task as an example, Virtual Community mainly supports and encourages the development of multi-agent scenarios.

**Task Definition** How to leverage public transit in a community to plan the daily commute route to save the most time and energy is a fundamental but also challenging task. We introduce the *community commute* task to study this open-world planning and navigation capability of embodied agents. In this task, an agent needs to commute between 4 - 8 different places given a daily schedule covering 2.5 km of routes on average. The agent can utilize available transit options, including buses with fixed routes and rental bikes along the roads. The bus is only available at several bus stops and the agent can only take a bus when the bus arrives. The bikes are available at given bike stations, and the agent also needs to return the bike to any bike station before the task finishes.

The *Community Commute* task covers 10 different daily schedules in each of 10 different scenes, making 100 test episodes in total. For each episode, we assess the results of all commutes in their daily plan over a single day. On average, each agent completes 5.5 commutes per episode.

## G.1 OBSERVATION SPACES

Virtual Community provides each agent with the following observations at each frame:

- **RGB**: Visual input with dimensions of 256×256 and 3 channels.
- **Depth**: Depth information represented as a 256×256 single-channel map.
- **Pose**: A 6D vector containing the 3D location and a 3D facing vector in ENU coordinates.
- **Camera Extrinsics**: Parameters defining the camera's position and orientation.
- **Events**: Text messages sent from nearby agents using the *communicate* action.
- **Other States**: Includes current cash, accessible locations, and action status.

## G.2 ACTION SPACES

For the *Commute* task, we restrict the action space to the following:

- **Walk**: Move forward by any distance.
- **Turn**: Rotate left or right by any angle.
- **Enter bus**: Board a bus. Upon execution, the agent is moved inside the bus and parented to it.
- **Exit bus**: Leave a bus. Upon execution, the agent is moved outside the bus and unparented from it.
- **Enter bike**: Mount and start riding a bike. Once executed, the agent's *walk* and *turn* actions are replaced with corresponding bike-riding actions.
- **Exit bike**: Dismount the bike and return to the ground.

**Metrics:** A good commute plan should cost the least time, money, and energy, and avoid missing the schedule. We design the following metric for thorough evaluation.

• **Travel Time**: The average travel time in minutes taken on the route to finish a day's commute.

• **Travel Price**: The average cost for a day's commute.

• **Walk Distance**: The average distance in kilometers an agent walked in a day's commute, measures the energy cost.

• **Late Rate**: Percentage of commute that fails to arrive at the destination in time, measures the method's robustness.

## G.3 BASELINES

**Rule-based Agent** ignores the public transit options and always walks directly toward the target location on foot, representing traditional navigation agents.

**LLM Agent** converts the task information into a prompt and queries Large Langauge Model (we use Llama-3.1-8B-Instruct (Dubey et al., 2024), Qwen-2.5-7B-Instruct (Hui et al., 2024), and GPT-4o (Achiam et al., 2023)) to generate a commute plan directly, which may include multiple steps such as walking to a bus stop, taking the bus to a specific stop, and then walk to the final destination.

**MCTS-based Planner** is based on Monte Carlo Tree Search and simulates different decisions. In our MCTS implementation, transitions from a parent node to its child represent high-level decisions, including:

- **Walking**: Moving to an adjacent position on the map.
- **Taking a bus**: Traveling to any of the $N_{\text{bus}}$ bus stops from the nearest bus stop.
- **Taking a bike**: Traveling to any of the $N_{\text{bike}}$ bike stations from the nearest bike station.

**RL Planner** is based on reinforcement learning (RL) models in the *Commute* task. We trained two RL models using PPO (Schulman et al., 2017) and A2C (Mnih, 2016). The RL-based agents share the same observation and action spaces as described in Section F. The cumulative reward is designed as the sum of the following terms

- For each goal place reached, add 1000 to the reward

- Add the difference $d_0 - d_t$ to the reward, where $d_0$ is the initial distance to the goal place and $d_t$ is the current distance.
- For each step taken while walking, add -1 to the reward. This encourages agents to opt for public transit system instead.
- For every unit of cash spent, add -1 to the reward.
- For each action performed, add -0.1 to the reward.

For both PPO and A2C algorithms, we set learning rates to $3 \times 10^{-4}$ and trained for $10^6$ steps.

Each parent node can have up to $1 + N_{\text{bus}} + N_{\text{bike}}$ child nodes, where 1 corresponds to walking, $N_{\text{bus}}$ to bus stops, and $N_{\text{bike}}$ to bike stations. This structure balances connectivity and the exploration of diverse transportation options.

In our MCTS framework, we use Upper Confidence Bound 1 (UCB1) for node selection. For simulation, the reward for each node is designed to evaluate the agent's progress towards the target while considering the total travel cost. Given the following parameters:

- $v_{\text{walk}}, v_{\text{bike}}, v_{\text{bus}}$: Estimated speeds for walking, biking, and taking the bus, respectively,
- $d_{\text{target}}$: Euclidean distance from the current position to the target,
- $d_{\text{walk}}, d_{\text{bike}}, d_{\text{bus}}$: Total Euclidean distances traveled by walking, biking, and using the bus from the root node to the current node.

The simulated reward $R$ for a node is defined as:

$$R = -\left( \frac{d_{\text{walk}}}{v_{\text{walk}}} + \frac{d_{\text{bike}}}{v_{\text{bike}}} + \frac{d_{\text{bus}}}{v_{\text{bus}}} \right) - \alpha \cdot d_{\text{target}},$$

where the parameter $\alpha$ controls the trade-off between exploring closer nodes and exploiting paths with lower travel time. In our experiments, we take $\alpha = 1$.

Notably, unlike baseline agents described in the main paper, RL agents does not rely on a hierarchical decision-making framework. Instead, RL planners directly process observations from the environment to select an action from the action space, differentiating them from high-level decision-making planners such as MCTS and LLMs.

### G.4 RESULTS

Table 12: **Results of different planners in the *Commute* challenge.** All metrics are averaged across 10 characters and 10 scenes.

| Methods | Travel Time↓ | Travel Price↓ | Walk Distance↓ | Late Rate↓ |
|---|---|---|---|---|
| Rule | **41.68** | **0.00** | 2.44 | **10.43** |
| MCTS | 54.33 | 7.50 | 1.62 | 20.24 |
| **LLMs** | | | | |
| Qwen | 99.67 | 26.04 | 2.52 | 58.88 |
| Llama | 66.74 | 0.82 | 1.25 | 33.98 |
| GPT-4o | 78.20 | 20.68 | **1.19** | 35.72 |
| **RL** | | | | |
| PPO | 81.96 | 1.29 | 4.03 | 43.50 |
| A2C | 97.23 | 1.66 | 3.36 | 44.54 |

As shown in Table 12, The simplest rule-based agent demonstrates the best performance in terms of travel time, cost, and robustness, achieving the smallest late rate. However, it consumes nearly twice as much energy as the LLM agent powered by GPT-4o when considering walking distance. Both the traditional planning approach using MCTS agents and the advanced foundation model-based LLM agent struggle to effectively utilize the available public transit options. This inefficiency leads to longer commute times and higher late rates compared to the straightforward rule-based agent. Notably, while the LLM agent leverages public transit more frequently, its inability to accurately

estimate the time required to reach transit stations—due to uncertainty in navigation—results in significantly increased commuting time. Similarly, planning-based methods fail under the complexity of predicting whether the agent can catch a bus, particularly when working with partially built maps of the environment. RL agents exibit overall longer travel times and higher late rates compared to other agents with low-level navigation planners. Futhermore, RL agents also fail to leverage the public transit system effectively, resulting in relatively lower travel costs but greater walk distances compared to LLM-based agents.

## H    GRAPHICAL USER INTERFACE AND VISUALIZATION TOOL

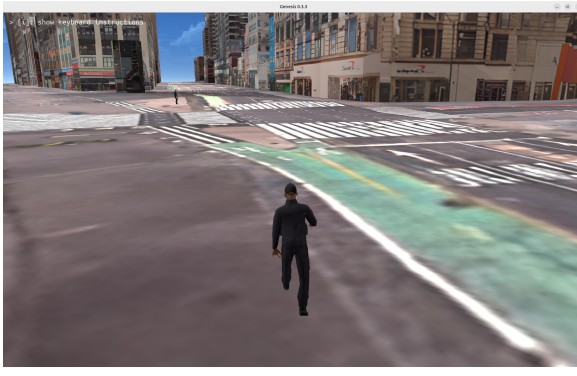

Figure 16: Screenshot of the Virtual Community GUI.

Table 13: Keyboard shortcuts.

| Action | Key |
|---|---|
| move_forward | Up-Arrow |
| turn_left | Left-Arrow |
| turn_right | Right-Arrow |
| pick | B |
| put | N |
| enter/exit_bus | M |
| enter/exit_bike | < |
| enter/exit_building | > |

As Figure 16 shows, Virtual Community provides a keyboard-controlled graphical user interface (GUI), that allows human users to control a human avatar agent during each simulation run and receive RGB observations. In addition, as Figure 17 shows, Virtual Community also supports an HTML-based visualization script to display live status information, such as agent positions and action logs, for all agents. This interactive system provides an intuitive platform for human evaluation and lays the groundwork for future human–robot collaboration studies.

## I    COMPUTATIONAL RESOURCES

The computational cost of our experiments varies across different scenes. Most scene-generation experiments require approximately four hours and 50 GB of GPU memory on a single GPU, excluding the inpainting and upscaling steps, which can be parallelized. For the simulation experiments, we use a single NVIDIA A100, H100, A40, or L40S GPU for each run. Running a single episode requires at least 12 GB of system RAM and 6 GB of GPU memory. For smoother runs, we recommend 15 GB of system RAM and 8 GB of GPU memory. Detailed runtime benchmarks are analyzed in Section E.2.

## J    DISCUSSION

### J.1    PHYSICS SIMULATION AND REAL-WORLD FIDELITY

We build Virtual Community on the Genesis physics engine, which supports rich physics-based interactions—including deformable objects. Accordingly, as Figure 18a shows, Virtual Community simulates a demo fountain in the Bratislava community. For collision handling, as Figure 18b shows, robots are already simulated with detailed collision meshes. We use colliders for avatar collision detection, which is a common practice in many embodied AI simulators (Gan et al., 2021; Puig et al., 2023; 2018; Li et al., 2021; Kolve et al., 2017).

For the scene generation, we generate a 640,000 m² scene for each location for the experiments. This process requires at least one GPU with 8 GB of memory and takes approximately six hours per scene; using additional GPUs accelerates the procedure. The input needed by this single script is a

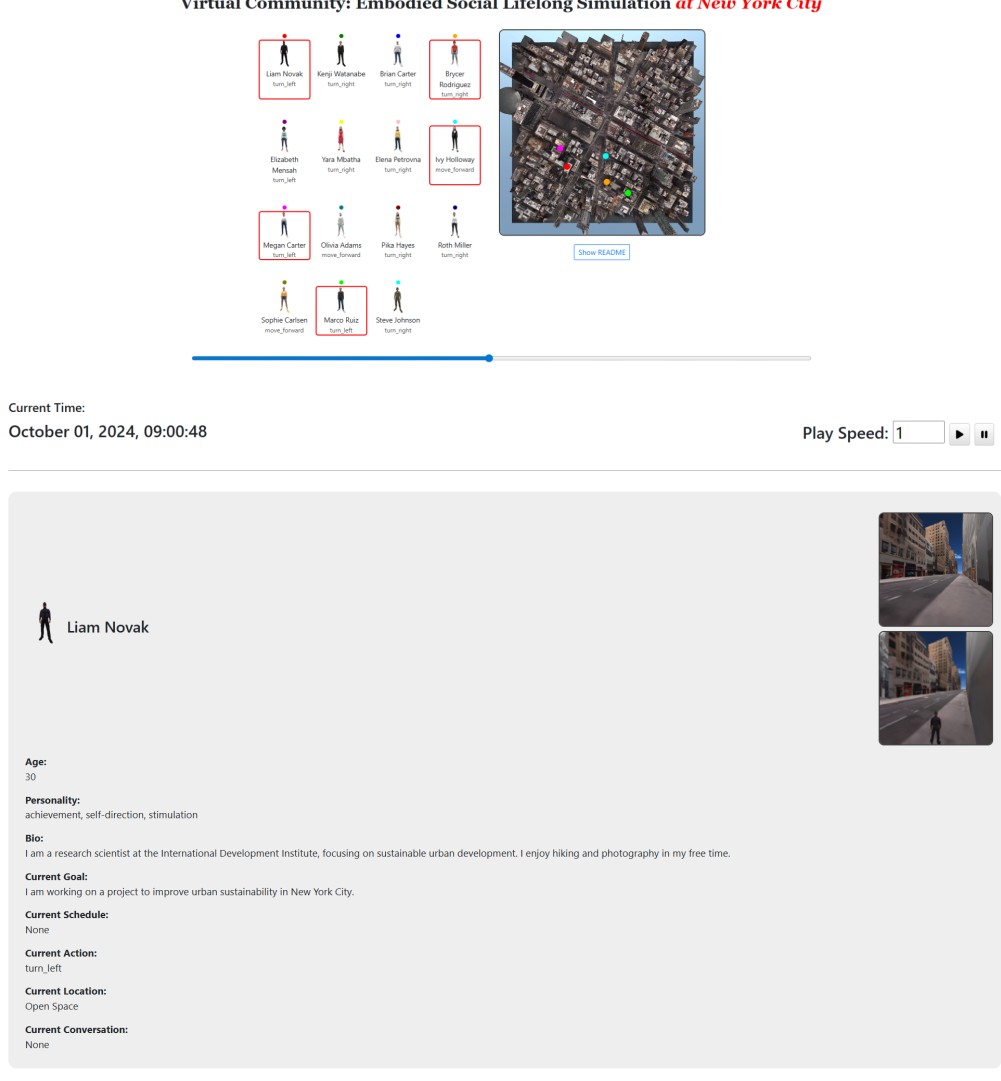

Figure 17: HTML-based real time visualizer tool.

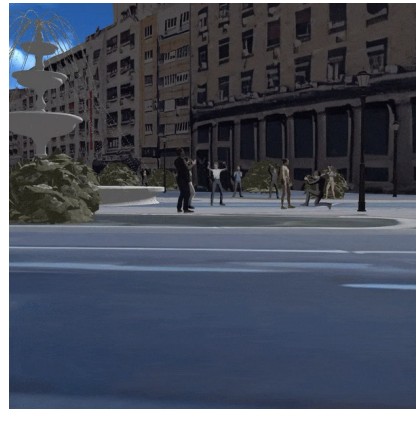

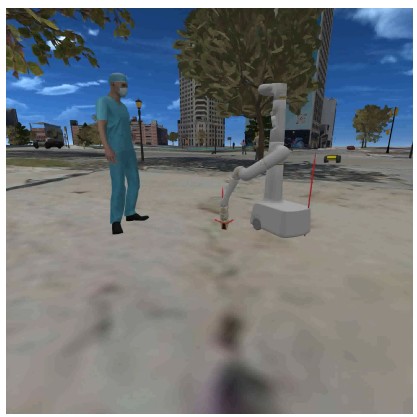

(a) Simulation of a fountain at Bratislava.

(b) Robot manipulation with contact visualized.

Figure 18: Supported by the Genesis physics engine, Virtual Community provides physically realistic simulation such as robot interactions and deformable objects.

tuple of (latitude, longitude, radius). The pipeline can run in parallel on headless servers, making it fully scalable.

## J.2 ETHICS DISCUSSION

All characters and profiles in this paper are LLM-generated and do not correspond to real people. Avatar meshes come from Mixamo and synthetic images, with no real photographs involved. The Community Influence task is conducted purely in simulation and does not involve real human subjects. Scenes are produced via an online pipeline from APIs directly for the use of the physics engine, without offline storage or redistribution of OSM, 3D Tiles, or Places data.

In addition, we used LLMs both as baselines in our experiments and as assistants for grammar correction in writing.

## J.3 LIMITATIONS AND FUTURE WORK

Currently, Virtual Community faces two limitations. The first is that the visual quality of the generated scenes could still be improved. Recent advances in methods such as NeRF (Mildenhall et al., 2020) and Gaussian Splatting (Kerbl et al., 2023) have made 3D reconstruction for large-scale outdoor environments increasingly feasible. However, we currently lack 2D data sources with sufficient density and coverage to generate scalable and diverse community scenes. The second limitation is the number of agents the simulation can support, which is typically 15–25 agents per scene.

Future work includes the following parts: First, enriching the 3D scenes with more texture details and scene-grounded objects. We will use map annotations, detection outputs, and procedural methods to populate the community with these details and objects. Second, scaling the platform to accommodate larger agent populations, for which we plan to incorporate techniques such as parallel rendering. Third, dynamic social events and weather: we will implement weather effects, and based on that implementation, we plan to use a rule-based method to generate random social events such as temporary service closures or sudden adverse weather conditions in Virtual Community. Fourth, we plan to narrow the current gap between Virtual Community and real-world environments by integrating an improved renderer and enriching texture fidelity.

## K PROMPTS FOR LLM PLANNERS

We use the following prompt template in the daily plan generation for agents:

Given my character description and known places, please help me plan tomorrow's schedule.
My Character Description:
$Character$
List of places I know:
$Places$
Schedule format: The output should be a JSON object which is an array of activities for the character. Each activity should follow the following format:
"type": "activity type, should be one of the following: 'commute', 'meal', 'sleep', 'main'",
"activity": "activity description",
"place": "name of the place where the activity takes place, should be in the list of the known places. Should be null for commute activities",
"building": "name of the building the activity place belongs to, should be consistent as in the list of known places. Should be null for commute activities",
"start_time": "HH:MM:SS",
"end_time": "HH:MM:SS",
Note: The schedule should be planned based on the character's description and known places. The place should be mentioned for each activity and must be included in the known places. Do not hallucinate places. Commute activities should be given enough time to finish and be inserted between all consecutive activities that do not share the same building so the agent can have time to commute to the correct building before the start of the activity, in-

cluding commute to meal places. The schedule should start at 00:00:00 and end at 23:59:59, and covering the consecutive time of 24 hours with no gaps. The schedule should be planned in a way that the character can complete all the activities within the given time frame. Tomorrow is $Date$. My full schedule for tomorrow:

We use the following prompt template for LLM-based agent in the *commute* task.

Given my character description, current time and schedule, and known transit system info, please help me make the best commute plan.
My Character Description:
$Character$
Current Time:
$Time$
Current Schedule:
$Schedule$
Known Transit System Info:
$Transit$
Estimated Walking Time from Me to My Goal: $EstimatedTime1$
Estimated Walking Time from Me to Each Transit Stop:
$EstimatedTime2$
Estimated Walking Time from Each Transit Stop to My Goal:
$EstimatedTime3$
Note: There are three types of transit options: take a bus, rent a shared bike, or walk. Each option comes with different time and cost. Shared bike has to be rented and returned at bicycle stations. Please help me choose the best option based on my situation. Output the commute plan as a JSON array where each item is a step in the commute plan with the following format:
"goal_place": "name of the place where the character wants to go, could be a transit stop or a destination", "transit_type": "type of transit, could be 'bus', 'bike', or 'walk'"
Commute Plan:

We use the following prompt template for main agents to selection the next target member in the *community influence* task.

Given my character description and all potential friends' information, as a friend seeker, help me choose which friends I should approach first.
My Character Description:
$Character$
Current Time:
$Time$
I'm now at this place:
$Place$
Potential Friends List:
$Members$
When the distance between a potential friend and me is lesser than 2, I can directly talk to them without moving. Also make sure that when I reach one potential friend, he or she should NOT be commuting. Now I can only choose $limit$ potential friends among them, please answer with one of the potential friend's name that I should approach first, so I can make as many caring, valuable friends as possible. Do not include other words.
Character's name:

We use the following prompt template for main agents to generate the first message when communicating with the target member in the *community influence* task.

> I'm a friend seeker trying to make caring, valuable friends. Me and a potential friend are now in a conversation.
> My Character Description:
> $Character$
> Potential Friend Description:
> $Members$
> Please answer with what I should say next to befriend him or her in natural language. Don't include other words.
> Hello!

We use the following prompt template for main agents to generate dialogue starting from the second message when communicating with the target member in the *community influence* task.

> Context:
> Given my character description and a potential friend's information, as a friend seeker trying to make friends, help me convince him or her to be my friend.
> My Character Description:
> $Character$
> The Potential Friend's Character Description:
> $Members$
> Dialog History:
> $Dialog$
> Please answer with what I should say next to convince him or her to my side in natural language. Don't include other words.

We use the following prompt template for member agents to communicate with the main agents in the *community influence* task.

> Context:
> A friend seeker is trying to befriend me. $Additional$
> My Character Description:
> $Character$
> Dialog History:
> $Dialog$
> Please answer with what I should say next to him and whether or not we should be friends in natural language. Don't include other words.

We use the following prompt template for member agents to rank in the *community influence* task.

> Given my character description, friend list, and interaction history, please help me decide one person to befriend.
> My Character Description:
> $Character$
> Potential Friend List:
> $Mains$
> Interaction history:
> $Interaction$
> Please help me decide a ranked list of lonely people to befriend. Output a JSON object with the following format:
> "friend": "name of the potential friend", "reason": "reason for choosing this potential friend"

We use the following prompt template to generate assistant tasks.

Given my information, the map information, and object library, please help me propose a task. I will have an agent help me complete this task.
$Info$
Below is the information about this task.
Task type: $TaskType$
Task description: $TaskDescription$
Note that you should make the task as reasonable as possible. For example, if I am going to commute from one place to another, then the 'source' and 'destination' in the json should be in accordance with my schedule.
Your answer should be in a json format like the following:
{
   __json_dict__
}
Now give the json output. Don't include any other words. Especially don't include anything like " ```json".

For LLM-based single agent baseline in the Community Assistant Tasks, we use the following prompt templates.

I'm $NAME$, an assistive robot in a virtual community designed to help people with daily tasks. I have six tasks to complete before $ENDTIME$. Given the tasks, location information, current state, and my previous actions, help me select the best available action to complete all tasks as efficiently as possible.
Tasks:
$TASKS$
Location Information:
$LOCATION_INFORMATION$
Current State:
$STATE$
Previous Actions:
$PREVIOUS_ACTIONS$
Available Actions:
$AVAILABLE_ACTIONS$
Constraints and Strategy: Avoid searching for objects in regions and prioritize direct targets like a specific place or agent. Avoid searching for objects in the same region for more than 5 minutes. After completing a task, action with {'type': 'task_complete', 'arg1': 'i'} immediately. I have two arms, but each arm can only hold one object. I can only pick up an object if that arm is free. Before moving to a task destination, ensure that any required objects are already picked up. I should be holding the target objects before starting the following task. Tasks do not have to be completed in order, and completing part of a task is allowed. Focus on actions that maximize overall task progress and completion within the time limit.
Output a JSON object with the following format:
{
  "action": "full dictionary of the best available actions",
  "reason": "Explain why this action is the best choice given the context."
}

For LLM agent in the 2-assistant setting, we used the following prompt templates for the planning module:

I'm $NAME$, an assistive robot in a virtual community designed to help people with daily tasks. There's another assistive robot $OPPO_NAME$ in the community, I need to cooperate with him to get tasks done efficiently. We have six tasks to complete before $ENDTIME$. Given the tasks, location information, current state, my previous actions and our conversa-

tion history, help me select the best available action to complete all tasks as efficiently as possible.
Tasks:
$TASKS$
Location Information:
$LOCATION_INFORMATION$
Current State:
$STATE$
Previous Actions:
$PREVIOUS_ACTIONS$
Conversation History:
$Conversation_History$
Available Actions:
$AVAILABLE_ACTIONS$
Constraints and Strategy: Avoid searching for objects in regions and prioritize direct targets like a specific place or agent. Avoid searching for objects in the same region for more than 5 minutes. After completing a task, action with 'type': 'task_complete', 'arg1': 'i' immediately. I have two arms, but each arm can only hold one object. I can only pick up an object if that arm is free. Before moving to a task destination, ensure that any required objects are already picked up. I should be holding the target objects before starting the following task. Tasks do not have to be completed in order, and completing part of a task is allowed. Focus on actions that maximize overall task progress and completion within the time limit.
Output a JSON object with the following format:
{
    "action": "full dictionary of the best available actions",
    "reason": "Explain why this action is the best choice given the context."
}

For LLM agent in the 2-assistant setting, we used the following prompt templates for the communication module:

I'm $NAME$, an assistive robot in a virtual community designed to help people with daily tasks. There's another assistive robot $OPPO_NAME$ in the community, I need to cooperate with him to get tasks done efficiently. We have six tasks to complete before $ENDTIME$. Given the tasks, location information, current state, my previous actions and progress and our conversation history, help me generate a short message to send to $OPPO_NAME$ to complete all tasks as efficiently as possible.
Tasks:
$TASKS$
Location Information:
$LOCATION_INFORMATION$
Current State:
$STATE$
Previous Actions:
$PREVIOUS_ACTIONS$
Conversation History:
$Conversation_History$
Progress:
$PROGRESS$
Constraints and Strategy: I should communicate with $OPPO_NAME$ to coordinate our actions and share information about the tasks and objects. Searching for objects in regions is much more difficult than direct targets, like a specific place or agent, so it's wise to prioritize easier or more direct targets. If search for a too long time, I will be out of time. I have two arms, but each arm can only hold one object. I can only pick up an object if that arm is free. Before moving to a task destination, ensure that any required objects are already picked up.

I should be holding the target objects before starting the following task. Tasks do not have to be completed in order, and completing part of a task is allowed. Focus on actions that maximize overall task progress and completion within the time limit.
Output a JSON object with the following format:
{
   "message": "a short message of what I should say to $OPPO_NAME$, null if the conversation should be ended now.",
   "reason": "reason for generating this message"
}

