# OpenReview forum: "Virtual Community: An Open World for Humans, Robots, and Society"
_ICLR.cc/2026/Conference — ICLR 2026 Poster_

### Official Review · Reviewer_xVwQ · 2025-10-31

**Soundness:** 4
**Presentation:** 4
**Contribution:** 4
**Rating:** 8
**Confidence:** 4

**Summary:**

This paper introduces Virtual Community, an open-world simulation platform for
studying interactions between multiple embodied agents. These embodied agents
can be either robots or humans. Virtual Community includes 3D open worlds where
multiple agents can interact with each other. Creating these worlds requires a
few steps. First is generating the world itself; this step starts with
geospatial data and then refines the data to create meshes that are easy to
simulate. The meshes are also photorealistic due to careful curation of
textures. The second step is to create the agents in the world; this is done by
prompting LLMs. Each agent has certain personalities and other attributes and
also a daily schedule of activities. Finally, all these agents and several
robots are simulated in the Genesis physics engine. To motivate studying
multiple embodied agents, this paper further introduces two challenges; the
Community Planning challenge studies social interactions among humans, while the
Community Robot Challenge studies how robots can interact with humans.

**Strengths:**

1. It is clear to me that the authors took a lot of time to polish this paper,
   as well as their accompanying website and codebase. In particular, I am
   impressed with the level of care taken to document the code and make it easy
   for folks to get started with Virtual Community.
1. The generation pipeline makes a lot of sense. In particular, I appreciate the
   use of LLMs to generate the agents automatically. It is nice that the
   resulting environment only requires a single GPU to run episodes.
1. I overall find the simulator quite novel and believe it provides a useful
   platform for studying embodied intelligence. I think the challenges are also a good
   starting point for the community; I am curious to see what methods can be created
   to solve the current challenges, and what new challenges can be created in the future.

**Weaknesses:**

I honestly could not find any major weaknesses with the paper. I have left
various minor comments and questions below.

**Questions:**

1. Line 241: "define their embodiments" -> "given their embodiments"?
1. I think it would be helpful to provide a line or two describing the Genesis
   engine, since Genesis is mentioned several times in the paper.
1. Section 4.1 mentions that the agents use a hierarchical planner. I am clear
   that the low-level actions are for navigating between locations. What are the
   high-level actions?
1. For the community assistant tasks, are the assistant agents robots or humans?
   Overall, it might be helpful to clarify in Section 4 whether "agent" refers
   to human or robot.
1. Do the authors anticipate creating more challenges within Virtual Community
   in the future?
1. I am curious whether the authors see any relation between this work and
   recent world modeling works like Genie 3?

---

> ### Author Response · Authors · 2025-11-26
> **Response to xVwQ**
>
> >Q1 Line 241: "define their embodiments" -> "given their embodiments"?
>
> Thanks! We updated it to “given their embodiments” for clarity.
>
> >Q2 I think it would be helpful to provide a line or two describing the Genesis engine, since Genesis is mentioned several times in the paper.
>
> Thank you for this constructive advice. We added a brief description of Genesis in Section 3.3. Genesis is a universal physics platform for general purpose embodied AI and robotics applications.
>
> >Q3 Section 4.1 mentions that the agents use a hierarchical planner. I am clear that the low-level actions are for navigating between locations. What are the high-level actions?
>
> Thank you for the suggestion. The high-level actions correspond to subgoal selection (e.g., navigating to a person or building), while low-level actions handle movement and object interactions. We have added this clarification in Section 4.1, and an example of high-level action set (excluding dialogue) was already provided in Fig. 14 in the Appendix.
>
> >Q4 For the community assistant tasks, are the assistant agents robots or humans? Overall, it might be helpful to clarify in Section 4 whether "agent" refers to human or robot.
>
> The community assistant tasks use human agents. Thank you for this valuable suggestion! we have updated the paper to consistently use “human agents” and “robot agents” to clearly distinguish the two throughout Section 4 and the rest of the text.
>
> >Q5 Do the authors anticipate creating more challenges within Virtual Community in the future?
>
> Yes. We will develop additional challenges in Virtual Community. Our current plan is that the environment can be initialized in multiple env modes along two main directions:
> 1. A social-focused env mode that simplifies part of the physics, scales up the number of agents, and introduces a richer event system.
> 2. A human–robot env mode with smaller-scale scenes but denser and more detailed outdoor objects, on top of which we will design new challenges centered on robot–human interaction.
>
> Thank you for the suggestion! we appreciate your valuable advice and welcome further feedback on potential challenge and designs.
>
> > Q6 I am curious whether the authors see any relation between this work and recent world modeling works like Genie 3?
>
> Thank you for this insightful question, it is indeed an interesting topic. Virtual Community targets at a *world simulator*, while works like Genie 3 focus on learning a *world model*. From this perspective, Genie 3–style world models and Virtual Community are complementary: Genie 3 learns powerful generative dynamics from data, while Virtual Community provides a geospatially grounded, physically realistic, and controllable training and evaluation platform for embodied social intelligence. In particular, a world simulator can supply world models with large-scale and diverse interaction data, making them more powerful and generalizable.

---

> > ### Comment · Reviewer_xVwQ · 2025-11-27
> >
> > I thank the authors for responding to my comments. I am happy to maintain my positive rating and outlook on this paper!

---

### Official Review · Reviewer_bndx · 2025-11-01

**Soundness:** 2
**Presentation:** 2
**Contribution:** 3
**Rating:** 6
**Confidence:** 4

**Summary:**

The goal of this paper is to introduce a simulation platform, built on top of the Genesis physics platform, that supports the simulation of urban spaces and populations of heterogeneous agents.


The contributions of this work are as follows:
1) Urban 3D scene generation pipeline.
2) Indoor 3D scene generation pipeline.
3) Multi-agent population generation pipeline.
4) Extension of Genesis to support humanoid agents.
5) Support for a large heterogeneous population of agents.
6) Multi-agent interaction tasks: Community Planning Challenge and The Community Robot Challenge.

**Strengths:**

1) Significance (human-robot interaction and embodied social intelligence are crucial research topics).
2) The work advances the capabilities for embodied social intelligence simulation and research.

**Weaknesses:**

1) The main weakness is work presentation quality (sentence formulation and writing style). While the paper's contributions are valuable, the clarity and style of the writing could be improved to reflect the quality of the underlying work better. A couple of comments:

- L33-36: Not clear how performed experiments (evaluation of baselines on introduced tasks) help to answer questions raised in the abstract: how robots cooperate or compete, how humans form social relations and communities and how humans and robots coexist. In other words, it feels like the these questions are not aligned with the rest of the paper. A more appropriate formulation would be "With Virtual Community, we aim to enable the study of embodied social intelligence at scale".


- L37-40: "A large-scale, real-world aligned community generation pipeline, including vast outdoor space, diverse indoor scenes, and a community of grounded agents with rich characters and appearances.". Community generation pipeline includes outdoor spaces, indoor spaces and community of agents? A better formulation might be "environment generation pipeline".

- Not consistent usage of terms in describing agents: L37: Virtual Community ... "supports robots, humans" agents. Then in L91-92 ...in Community Planning challenge ... "agents interact with humans and other agents". Then L300-301: community assistant tasks "in which agents cooperatively plan to assist multiple humans". And L301 community influence task "in which agents competitively plan to efficiently connect and interact with other agents". This makes it confusing to understand what types of agents are supposed to be involved in a particular task.

- Another semantically overloaded phrase is open world. "Open-world" - used 30 times, "open world" - used 19 times. However, it is not explicitly defined what it means in the context of the work.

2) It is hard to draw insights about embodied social intelligence from selected baselines and performed experiments. Tables demonstrate which baselines perform better compared to other baselines.

**Questions:**

1) Grounding Validator performs scene grounding validation (checks whether generated agent profile matches scene). What about validation of generated agent profiles and community relations, to make sure they reflect real worls scenarios?

2) What is the maximum size of the population that Virtual Community supports?

3) L83: Remove dot.

4) L1069-1073 repeats L1098-1103.

5) L1208-1209: "We utilize the according URDF file" -> "We utilize the corresponding URDF files"?

---

> ### Author Response · Authors · 2025-11-26
> **Response to bndx**
>
> >Q1 L33-36: Not clear how performed experiments (evaluation of baselines on introduced tasks) help to answer questions raised in the abstract: how robots cooperate or compete, how humans form social relations and communities and how humans and robots coexist. In other words, it feels like the these questions are not aligned with the rest of the paper. A more appropriate formulation would be "With Virtual Community, we aim to enable the study of embodied social intelligence at scale".
>
> Thank you for the suggestion. We completely agree and have revised it to "enabling the study of embodied social intelligence at scale", which better aligns with the experiments and scope of the paper.
>
> >Q2 L37-40: "A large-scale, real-world aligned community generation pipeline, including vast outdoor space, diverse indoor scenes, and a community of grounded agents with rich characters and appearances.". Community generation pipeline includes outdoor spaces, indoor spaces and community of agents? A better formulation might be "environment generation pipeline".
>
> Thank you for the suggestion. We agree that “environment generation pipeline” is clearer. We have updated the wording accordingly in the revision.
>
> >Q3 Not consistent usage of terms in describing agents: L37: Virtual Community ... "supports robots, humans" agents. Then in L91-92 ...in Community Planning challenge ... "agents interact with humans and other agents". Then L300-301: community assistant tasks "in which agents cooperatively plan to assist multiple humans". And L301 community influence task "in which agents competitively plan to efficiently connect and interact with other agents". This makes it confusing to understand what types of agents are supposed to be involved in a particular task.
>
> Thanks for the feedback. We have standardized the terminology: “agents” is used as the generic term, while “robot agents” and “human agents” are explicitly distinguished when needed. For each task, we now clearly specify which types of agents are involved, removing the previous ambiguity.
>
> >Q4 Another semantically overloaded phrase is open world. "Open-world" - used 30 times, "open world" - used 19 times. However, it is not explicitly defined what it means in the context of the work.
>
> Thanks for pointing this out. We now define “open-world” as where an agent can freely explore a large, non-linear world instead of being restricted to a fixed path or sequence of levels. We added this definition to the first paragraph of the Introduction Section in the revision.
>
> >Q5 It is hard to draw insights about embodied social intelligence from selected baselines and performed experiments. Tables demonstrate which baselines perform better compared to other baselines.
>
> Thank you for the comment. Beyond comparing baseline scores, our experiments reveal why embodied social intelligence is challenging.
> 1. In the community assistant tasks, different baselines fail for different reasons (e.g., LLMs struggle with tracking task progress, while heuristics mis-handle open-world costs and fail to cooperate effectively), highlighting limits in task-aware cooperative planning.
> 2. In the influence task, stronger LLMs consistently choose better social targets and produce more persuasive dialogues, showing clear differences in social-reasoning ability.
> 3. Together, these tasks show distinct aspects of embodied social intelligence, and the benchmark supports evaluating new methods on these aspects.
>
>
> >Q6 Grounding Validator performs scene grounding validation (checks whether generated agent profile matches scene). What about validation of generated agent profiles and community relations, to make sure they reflect real worls scenarios?
>
> Thank you for the valuable question. Scene–profile validation is performed by the Grounding Validator. As for the realism of social relations and character profiles, this is partially addressed through our character and social-network generation pipeline. In this pipeline, the LLM generates profiles conditioned on manually verified attributes such as age and appearance. So that the resulting personalities and relationships remain coherent and realistic.
>
> >Q7 What is the maximum size of the population that Virtual Community supports?
>
> That is a good question. The upper limit primarily depends on hardware capacity. In our tests, we have run up to 25 agents. Most experiments use 15 agents. We are currently doing a memory benchmarking and will provide a detailed table soon.
>
> >Q8 Grammar:
> L83: Remove dot.
> L1069-1073 repeats L1098-1103.
> L1208-1209: "We utilize the according URDF file" -> "We utilize the corresponding URDF files"?
>
> Thanks for the detailed and actionable suggestions. All issues have been fixed in the updated paper.

---

> ### Author Response · Authors · 2025-12-01
> **Additional Response to bndx**
>
> >Q7 What is the maximum size of the population that Virtual Community supports?
>
> Additional response: In our memory tests, we run up to 50 agents in a single process with all scene assets loaded. We conduct the memory benchmarking on a single NVIDIA L40S GPU machine. The results are summarized below.
>
> | Num. agents | Peak RAM (GB) | Avg. RAM (GB) | Peak GPU (GB) |
> |---:|---:|---:|---:|
> | 1  | 17.45 | 15.11 | 4.38 |
> | 5  | 18.81 | 16.94 | 4.88 |
> | 10 | 21.14 | 19.28 | 6.00 |
> | 15 | 23.47 | 21.58 | 7.02 |
> | 20 | 25.87 | 23.95 | 8.05 |
> | 25 | 28.22 | 26.08 | 9.16 |
> | 30 | 30.46 | 28.39 | 10.16 |
> | 35 | 33.30 | 30.24 | 10.65 |
> | 40 | 33.90 | 31.69 | 12.26 |
> | 45 | 36.29 | 34.17 | 13.24 |
> | 50 | 38.46 | 36.01 | 14.26 |
>
> Overall, memory usage scales approximately linearly with the number of agents. From 1 to 50 agents, the memory footprint increases by about 0.43 GB peak RAM, 0.43 GB average RAM, and 0.20 GB peak GPU memory per additional agent, indicating stable and predictable memory scaling for large agent communities.
>
> Thank you for the question and we have added this part into the paper Appendix E.2.

---

### Official Review · Reviewer_5oy8 · 2025-11-03

**Soundness:** 3
**Presentation:** 3
**Contribution:** 4
**Rating:** 8
**Confidence:** 4

**Summary:**

The paper presents Virtual Community, a city-scale simulation platform that unifies human avatars, multiple robot types, and social structure (profiles, schedules, relationships) within a single physics-enabled engine. It proposes two families of benchmarks: Community Planning (carry/delivery/search and a social-influence setting) and Community Robot (heterogeneous multi-robot mobile manipulation). The system couples real-world geographic data (OSM/Google) with a generative asset pipeline to produce large outdoor areas, indoor scenes, and populated communities. Baselines include heuristic/MCTS/LLM planners and IK+RRT vs. PPO for manipulation.

The work is ambitious and likely to be useful to the community, but several claims and evaluations need tightening (especially around “physically real” and manipulation without oracle priors).

**Strengths:**

1. Scope & Unification – A rare integration of city-scale environments, human–robot co-presence, and social graphs with time-aligned schedules, enabling questions that neither indoor-only simulators nor purely social agents can address.
2. Real-world Grounding at Scale – Uses real geographic data and a generative pipeline to create semantically rich, traversable outdoor/indoor spaces (transport, buildings, POIs). This improves task relevance for long-horizon planning.
3. Well-defined Benchmarks – Clear task families (carry/delivery/search/influence) with consistent observations/actions and success/time/follow metrics; results reveal non-trivial failure modes for LLM planners and for manipulation.
4. Transparent Baselines – A modular navigation stack, multiple planners (Random/Heuristic/MCTS/LLM), and classical IK+RRT vs. RL for manipulation provide a solid starting point and set realistic expectations.
5. Community Value – the platform can serve as a bridge between embodied AI and social simulation, attracting researchers from planning, VLA, multi-agent RL, and HRI.

**Weaknesses:**

1. Over-claim on “physical realism.”
The implementation appears primarily rigid-body + contact with kinematic attach/detach for human–object/vehicle interactions. There is limited validation of compliant contact, actuator dynamics, deformables/fluids, or grasp stability. The current evidence supports physics-enabled rather than physically real.

2. Manipulation relies on oracle priors; RL underperforms.
Success rates drop notably without oracle grasps; end-to-end RL struggles in sparse, long-horizon settings. This weakens the claim that the platform currently supports robust mobile manipulation in open worlds.

3. State/Progress tracking in LLM planning.
LLM planners excel at search but degrade on multi-step, progress-dependent tasks (carry/delivery). Lack of explicit memory/plan monitoring leads to mis-ordering and cost underestimation.

4. Limited quantitative validation of physics and assets.
The paper acknowledges outdoor detail gaps but does not provide measurable physical consistency tests (e.g., friction/solver sensitivity) or cross-sim comparisons to indoor high-fidelity platforms.

5. Navigation & scheduling cost modeling.
While transit/indoor-outdoor transitions are supported, there is no ablation on commute cost modeling (wait times, congestion, transfers) or task ordering policies, which likely drive failures in open-world itineraries.

**Questions:**

1. What exact physics features are enabled at run-time (solver, time step, iterations, contact model, friction model), and how sensitive are benchmark outcomes to them?

2. How are kinematic attachments triggered/terminated, and do they bias success metrics versus true closed-loop grasp stability?

3. Can the authors provide statistics on map connectivity and accessibility (e.g., average indoor-outdoor traversal times, transit wait distributions) and correlate them with failures?

4。 For the influence task, how are dialogue vs. target-selection effects disentangled? Any evaluation with stronger memory or planning scaffolds?

---

> ### Author Response · Authors · 2025-11-26
> **Response to 5oy8 Part 1**
>
> >Q1 Over-claim on “physical realism.”
>
> Thanks for the feedback. Our platform is designed for embodied AI research with both robotics and avatars. Since low-level robot control remains an open problem, our design allows researchers to study high-level reasoning and cooperation without being constrained by low-level control policies. For robotics studies that require physical realism, because Virtual Community is built directly on Genesis, all of Genesis’s physical modeling capabilities and validations naturally carry over to our system. We address the reviewer’s concerns point by point by summarizing the relevant properties of Genesis:
>
> * Compliant Contact: Within its rigid-body solver, Genesis uses GJK (Gilbert-Johnson-Keerthi) and MPR (Minkowski Portal Refinement) algorithms to detect collisions, and adopts MuJoCo’s contact model (https://ieeexplore.ieee.org/document/6386109
> ), a quadratic-optimization–based compliant contact formulation. This model is one of the most widely used solvers in robotics [1,2,3,4] and has been extensively validated across both simulation and real-world systems.
> * Actuator Dynamics: Genesis includes MuJoCo-style control algorithms, and with ongoing updates, incorporates even more accurate actuator models—such as the recent pull request:
> https://github.com/Genesis-Embodied-AI/Genesis/pull/1948
>  The Virtual Community built on Genesis will automatically inherit these improvements.
> * Deformable Simulation: Genesis supports simulating a wide range of materials and physical phenomena, including continuum methods such as Material Point Method (MPM)[5] for deformable objects, and particle-based solvers like Smoothed Particle Hydrodynamics (SPH)[6,7] for fluids. It also integrates state-of-the-art coupling solvers acorss different materials, including the Incremental Potential Contact (IPC)[8] as well as Drake’s SAP solver combined with the hydroelastic contact model [9].
> * Grasp Stability: Genesis’s grasp stability has also been validated, for example in the DexMachina project[10].
>
> [1] OpenAI Gym. Brockman et al., arXiv 2016.
>
> [2] DeepMind Control Suite. Tassa et al., arXiv 2018.
>
> [3] Robosuite: A Modular Simulation Framework and Benchmark for Robot Learning. Zhu et al., arXiv 2020.
>
> [4] Meta-World: A Benchmark and Evaluation for Multi-Task and Meta Reinforcement Learning. Yu et al., PMLR 2020.
>
> [5] A Particle Method for History-Dependent Materials. Sulsky et al., Computer methods in applied mechanics and engineering 1994.
>
> [6] Smoothed particle hydrodynamics: theory and application to non-spherical stars. Gingold et al., MNRAS 1977.
>
> [7] A numerical approach to the testing of the fission hypothesis. Lucy, et al., The Astronomical Journal 1977.
>
> [8] Incremental potential contact: intersection-and inversion-free, large-deformation dynamics. Li et al., ACM Trans. Graph 2020.
>
> [9] An Unconstrained Convex Formulation of Compliant Contact. Castro et al., IEEE Transactions on Robotics, 2022.
>
> [10] DexMachina: Functional Retargeting for Bimanual Dexterous Manipulation. Zhao et al., arXiv 2025.
>
> >Q2 Manipulation relies on oracle priors; RL underperforms. Success rates drop notably without oracle grasps; end-to-end RL struggles in sparse, long-horizon settings.
>
> Thanks for raising this concern. First, our platform is primarily designed to study long-horizon, open-world tasks, where sparse rewards and extended action sequences are inherently challenging. It is common that standard end-to-end RL methods tend to underperform in such settings [1,2,3,4], so the lower success rates are expected rather than evidence that the platform cannot support robust mobile manipulation.
>
> Second, to better isolate the difficulty, we followed the reviewer’s suggestion and further decomposed the manipulation pipeline. Specifically, we split the RL trained on the original pick-and-place task into two smaller subtasks, and implemented a new baseline: a decomposed RL agent. The results are as follows:
>
> | **Method**                | **Carry SR↑** | **Carry Ts↓** | **Deliver SR↑** | **Deliver Ts↓** | **Avg SR↑** |
> |---------------------------|---------------|---------------|------------------|------------------|-------------|
> | RL w/ Oracle Grasp        | 19.0          | 149.7         | 42.9             | 168.1            | 31.0        |
> | RL Decomposed w Oracle Grasp        | 19.0          |   149.0       | 47.6             |       172.1      | 33.3        |
>
> The decomposed variant slightly outperforms the original RL baseline. We have added this table to Section 5.
>
> [1] Building Cooperative Embodied Agents Modularly with Large Language Models. Zhang et al., ICLR 2024.
>
> [2] Auxiliary Tasks and Exploration Enable ObjectNav. Ye et al., ICCV 2021.
>
> [3] Exploration in Deep Reinforcement Learning: A Survey. Ladosz et al., arXiv 2022.
>
> [4] Hierarchical Reinforcement Learning: A Survey and Open Research Challenges. Hutsebaut-Buysse et al., Machine Learning and Knowledge 2022.

---

> ### Author Response · Authors · 2025-11-26
> **Response to 5oy8 Part 2**
>
> >Q3 State/Progress tracking in LLM planning. LLM planners excel at search but degrade on multi-step, progress-dependent tasks (carry/delivery). Lack of explicit memory/plan monitoring leads to mis-ordering and cost underestimation.
>
> Thank you for the insightful question. How to track state and progress in LLM-based planners is indeed an important open problem. As discussed in L370, we adopt CoELA’s design, which explicitly monitors task progress throughout execution. Our experimental results show that the proposed tasks are genuinely challenging and can serve as a rigorous benchmark for studying state/progress tracking in LLM planners and advancing research in this direction.
>
> >Q4 Limited quantitative validation of physics and assets. The paper acknowledges outdoor detail gaps but does not provide measurable physical consistency tests (e.g., friction/solver sensitivity) or cross-sim comparisons to indoor high-fidelity platforms.
>
> Thanks for the question. We provided quantitative analysis of outdoor assets in Appendix A.5. Physics parameters discussion are included in our response to Q6, and simulator-level physics consistency is included in our answer to Q1.
>
> >Q5 Navigation & scheduling cost modeling. While transit/indoor-outdoor transitions are supported, there is no ablation on commute cost modeling (wait times, congestion, transfers) or task ordering policies, which likely drive failures in open-world itineraries.
>
> Thank you for the valuable question. Following your suggestion, we add ablation studies on the commute cost modeling for both the LLM agent and the MCTS agent. Since the heuristic agent does not model commute cost, we did not include it in this ablation. The results are shown below
>
> | Method | Carry SR↑ | Carry HR↑ | Delivery SR↑ | Delivery TS↓ | Search SR↑ | Search TS↓ |
> |--------|-----------|-----------|---------------|--------------|---------------|--------------|
> | 1_no_pos_mcts | 33.3 | 7.0 | 29.9 | 1500.0 | 23.9 | 1500.0 |
> | 1_no_pos_llm | 27.5 | 14.6 | 39.7 | 1500.0 | 66.0 | 1236.2 |
> | 2_no_pos_mcts | 34.0 | 6.9 | 27.1 | 1500.0 | 27.1 | 1500.0 |
> | 2_no_pos_llm | 25.0 | 15.2 | 55.6 | 1432.9 | 76.4 | 1060.2 |
>
> The results indicate that the MCTS Agent relies far more heavily on accurate distance modeling to perform the task effectively. In contrast, although the LLM Agent also exhibits a notable performance drop in the 1-assistant setting, in the 2-assistant setting it can sometimes even surpass its original baseline. This suggests that cooperative interaction enables the LLM Agent to partially compensate for the absence of explicit distance modeling.
>
> As for task ordering, this behavior is determined by each baseline method’s inherent decision-making mechanism. For example, the LLM agent selects the task execution order directly through LLM reasoning, while the MCTS agent determines the order through tree search to optimize long-horizon returns. The results and the analysis are updated in **Section 4.1.**
>
> >Q6 What exact physics features are enabled at run-time (solver, time step, iterations, contact model, friction model), and how sensitive are benchmark outcomes to them?
>
> Thank you for the question.
> * We run with a fixed time step of 1e−2, using conjugate gradient constraint solver with Euler integrator. Contact and friction model follow the Genesis defaults, and all methods share the same physics settings.
> * Solver/contact-model sensitivity is an important research problem, but it is out of scope for this paper.
> * We adopt Genesis’s best available, real-world-oriented default solver stack to provide the most appropriate and stable testing environment for all baselines, and we evaluate every method under identical physics settings for a controlled comparison.
> * This sensitivity largely belongs to the Genesis physics engine layer. As Computer Graphics community rapidly progress, simulation stability continues to improve.

---

> ### Author Response · Authors · 2025-11-26
> **Response to 5oy8 Part 3**
>
> >Q7 How are kinematic attachments triggered/terminated, and do they bias success metrics versus true closed-loop grasp stability?
>
> Thanks for highlighting this important point. Kinematic attachment (oracle grasping) in our simulator is triggered purely based on Euclidean distance: when the object’s center is within a distance threshold (we use 0.05m) of the gripper’s center the object is attached. This follows the kinematic pickup / grasp implementation used in existing simulators.
>
> We acknowledge that such kinematic attachment can influence grasp success metrics. For this reason, our oracle-grasp RL baseline is trained and evaluated under the same oracle setting, ensuring consistency and avoiding bias.
>
> To address the reviewer’s concern directly, we additionally trained an RL agent without kinematic attachment. The results are reported below:
>
> | **Method**                | **Carry SR↑** | **Carry Ts↓** | **Deliver SR↑** | **Deliver Ts↓** | **Avg SR↑** |
> |---------------------------|---------------|---------------|------------------|------------------|-------------|
> | Heuristic w/ Oracle Grasp | **23.5**      | **124.4**     | **50.0**         | 131.2            | **36.8**    |
> | Heuristic  w/o Oracle Grasp      | 17.6          | 126.9         | 22.2             | **129.4**        | 19.9        |
> | RL w/ Oracle Grasp        | 19.0          | 149.7         | 42.9             | 168.1            | 31.0        |
> | RL w/o Oracle Grasp        | 9.5          | 143.6         | 19.0             | 166.7            |    14.3     |
>
> The results show that RL without Oracle Grasp struggles in both tasks, which is consistent with the pattern in the heuristic baselines. We have added these results to Section 5.
>
> >Q8 Can the authors provide statistics on map connectivity and accessibility (e.g., average indoor-outdoor traversal times, transit wait distributions) and correlate them with failures?
>
> Your concern regarding map connectivity is valuable, and we appreciate the opportunity to clarify this. We have previously examined this issue in detail. In our anonymous codebase, the function is_point_enclosed in Virtual-Community/tools/utils.py is specifically designed to detect whether a point is enclosed or disconnected, using a flood-fill-like algorithm. During agent initialization, we use this function to verify every candidate starting position and reject any point that is not connected to the map boundary. This guarantees global reachability for all initialized agents. We have added this description to Appendix B.2.
>
> In addition, in the single-agent commute challenge, we systematically evaluated the travel time between arbitrary points on the map, represented through commute cost, success rate, and step count. This further confirms that connectivity does not pose a failure factor.
>
>
> >Q9 For the influence task, how are dialogue vs. target-selection effects disentangled? Any evaluation with stronger memory or planning scaffolds?
>
> This is a good point. To address it, we ran ablations that disentangle the effects of dialogue and target selection by swapping each component with different models while keep the others fixed. The results are listed below.
>
> |Dialogue| Target Selection | Success Rate | Conversion Rate|
> |--|--|--|--|
> |gpt-4o|gpt-4o|0.57|0.17|
> |gpt-4o|o1|0.58|0.06|
> |o1|gpt-4o|0.60|**0.20**|
> |o1|o1|**0.63**|0.13|
>
> The results indicate that both target selection and dialogue generation improve with a stronger backbone, while dialogue generation contributes more substantially to the overall performance gains.

---

### Author Response · Authors · 2025-11-26
**General Response by Authors**

Thank you to all reviewers for the thoughtful and constructive feedback, and for acknowledging the strengths and potential impact of Virtual Community [5oy8, bndx, xVwQ]. We particularly appreciate the concrete guidance on where the paper can be improved.

Virtual Community features its scope in supporting city-scale environments [5oy8, bndx, xVwQ], its real-world grounding [5oy8], and its unified simulation of humans and robots within a single platform [5oy8, bndx, xVwQ]. It also provides well-defined challenges/benchmarks [5oy8], along with detailed documentation and code [xVwQ].

We have updated the paper based on your constructive suggestions, and all changes are highlighted in red in the revised paper. Thank you again for your time and valuable feedback.

---

### Meta-Review · Area_Chair_5Ubs · 2025-12-26

**Summary:**

The paper introduces Virtual Community, a city-scale simulation platform built on the Genesis physics engine for studying embodied social intelligence. It features a generative pipeline for real-world 3D environments and populates them with heterogeneous agents (robots and humans) with rich characters and daily schedules. The authors propose two benchmarks: Community Planning and Community Robot challenges. Reviewers consistently lauded the platform's scale, unification of social/physical simulation, and its potential value as a bridge between embodied AI and social reasoning.

**Reviewer Concerns:**

Addressed in Rebuttal:
- Physical Realism: Authors clarified the underlying physics capabilities of Genesis (compliant contact, actuator dynamics) and moved from "physically real" to "physics-enabled".
- Manipulation & Oracle Priors: New experiments were provided for RL agents without oracle grasps and decomposed tasks, setting realistic expectations for current baseline performance.
- Terminology & Presentation: Concerns regarding the inconsistent use of "agents" and "open world" were addressed through standardized definitions and text revisions.
- Scaling: Memory benchmarking was added, demonstrating stable linear scaling up to 50 agents on a single GPU.

Outstanding Concerns:
None that significantly impact the recommendation for acceptance; reviewers expressed satisfaction with the comprehensive responses

**Reviewer Scores:**

Reviewers are likely to stay on the positive side.

---

### Decision · Program_Chairs · 2026-01-26

Accept (Poster)